# Dendritic Na⁺ spikes enable cortical input to drive action potential output from hippocampal CA2 pyramidal neurons

Qian Sun[1]*, Kalyan V Srinivas[1], Alaba Sotayo[1], Steven A Siegelbaum[1,2]*

[1]Department of Neuroscience, Howard Hughes Medical Institute, Columbia University, New York, United States; [2]Department of Pharmacology, Kavli Institute for Brain Science, Columbia University, New York, United States

**Abstract** Synaptic inputs from different brain areas are often targeted to distinct regions of neuronal dendritic arbors. Inputs to proximal dendrites usually produce large somatic EPSPs that efficiently trigger action potential (AP) output, whereas inputs to distal dendrites are greatly attenuated and may largely modulate AP output. In contrast to most other cortical and hippocampal neurons, hippocampal CA2 pyramidal neurons show unusually strong excitation by their distal dendritic inputs from entorhinal cortex (EC). In this study, we demonstrate that the ability of these EC inputs to drive CA2 AP output requires the firing of local dendritic Na⁺ spikes. Furthermore, we find that CA2 dendritic geometry contributes to the efficient coupling of dendritic Na⁺ spikes to AP output. These results provide a striking example of how dendritic spikes enable direct cortical inputs to overcome unfavorable distal synaptic locale to trigger axonal AP output and thereby enable efficient cortico-hippocampal information flow.

## Introduction

The active properties of neuronal dendrites are important for integrating and processing excitatory and inhibitory synaptic inputs (*Johnston et al., 1996*; *London and Hausser, 2005*; *Johnston and Narayanan, 2008*; *Major et al., 2013*). Over the past few decades, dendritically generated Na⁺, Ca²⁺, and NMDA spikes have been identified in many types of neurons, both in vitro and in vivo (*Llinas et al., 1968*; *Wong et al., 1979*; *Stuart and Sakmann, 1994*; *Chen et al., 1997*; *Schiller et al., 1997, 2000*; *Stuart et al., 1997a*; *Kamondi et al., 1998*; *Larkum et al., 1999*; *Waters et al., 2003*; *Larkum et al., 2007, 2009*; *Kim et al., 2012*; *Smith et al., 2013*). One proposed function of dendritic Na⁺ spikes is to amplify synaptic potentials and facilitate somatic AP initiation (*Hausser et al., 2000*; *London and Hausser, 2005*). However, in most instances, dendritic Na⁺ spikes propagate poorly to the soma and so fail to act as reliable triggers of somatic APs (*Stuart and Sakmann, 1994*; *Stuart et al., 1997a*; *Golding and Spruston, 1998*). Indeed, under physiological conditions, the APs in most principal neurons, including neocortical layer 5 and hippocampal CA1 pyramidal neurons (PNs), are usually initiated at the axonal initial segment (AIS) before back-propagating to the dendrites (*Stuart and Sakmann, 1994*; *Stuart et al., 1997a*, *1997b*; *Golding and Spruston, 1998*). Thus, whereas dendritic Na⁺ spikes can fine-tune neuronal output and regulate synaptic plasticity (*Golding et al., 2002*; *Ariav et al., 2003*; *Jarsky et al., 2005*; *Remy and Spruston, 2007*), it is less certain whether these spikes may serve as necessary events to allow synaptic input to trigger AP output (*Hausser et al., 2000*; *Spruston, 2008*). In this study, we report that dendritic Na⁺ spikes play an important role in the ability of hippocampal CA2 PNs to generate axonal AP output in response to synaptic input from the direct entorhinal cortical (EC) projections that terminate on CA2 PN distal dendrites.

Hippocampal CA2 PNs represent a relatively small population of cells interspersed between CA3 and CA1. Nonetheless, these neurons have recently been shown to be crucial for social memory (*Hitti and Siegelbaum, 2014*; *Stevenson and Caldwell, 2014*) and aggression (*Pagani et al., 2014*).

*For correspondence: qsun79@ gmail.com (QS); sas8@columbia. edu (SAS)

**Competing interests:** The authors declare that no competing interests exist.

**Reviewing editor**: Gary L Westbrook, Vollum Institute, United States

**eLife digest** Cells called neurons carry information—in the form of electrical signals—around the brain. These cells connect to each other in complex networks and each neuron is able to form junctions, or synapses, with many neighbors. In a neuron, small electrical signals start from synapses at the tips of branched structures called dendrites. From there, these signals travel to the cell body of the neuron to activate a larger electrical signal—called an action potential—that travels along a long tail-like extension, called the axon, to reach synapses with other neurons.

In the dendrites, the small electrical signals can be amplified by rapid changes in the concentration of sodium ions, known as $Na^+$ spikes. Although they were first recorded over 40 years ago, it is not clear how important the $Na^+$ spikes are for triggering action potentials.

In this study, Sun et al. studied a type of neuron in the hippocampus called CA2 pyramidal neurons, which are involved in social memory and aggression. Unlike most other neurons in this region, CA2 neurons are strongly activated by signals from a neighboring region of the brain called the entorhinal cortex. The experiments show that $Na^+$ spikes are able to travel from the dendrites to the cell body of these neurons, where they are required to trigger action potentials. However, this is not the case for other neurons in the hippocampus, where the $Na^+$ spikes are very weak by the time they reach the cell body.

Sun et al. used a computational modeling technique to compare the different types of neurons in the hippocampus. The dendrites of these cells have different branching patterns and shapes, and the model suggests that this may explain the differences in how well the $Na^+$ spikes travel to the cell body. The next major challenge is to understand the role of the $Na^+$ spikes in social memory and other complex behaviors that are controlled by CA2 neurons.

CA2 PNs also have unique synaptic properties that distinguish them from their CA1 and CA3 neighbors (*Zhao et al., 2007*; *DeVito et al., 2009*; *Simons et al., 2009*; *Chevaleyre and Siegelbaum, 2010*; *Lee et al., 2010*; *Jones and McHugh, 2011*; *Caruana et al., 2012*; *Hitti and Siegelbaum, 2014*). Thus, whereas the perforant path (PP) inputs from the EC form weak excitatory synaptic connections at the distal dendrites of CA1 PNs located in stratum lacunosum-moleculare (SLM), PP inputs to CA2 PNs provide a much stronger excitatory drive (*Chevaleyre and Siegelbaum, 2010*). In contrast the Schaffer collateral (SC) inputs to CA2 are relatively weak and dominated by powerful feed-forward inhibition (*Chevaleyre and Siegelbaum, 2010*). Finally, individual CA2 PNs provide stronger excitatory drive to CA1 compared to the weaker influence of single CA3 SC inputs (*Chevaleyre and Siegelbaum, 2010*). Such properties enable the CA2 region to function as the nexus of a powerful disynaptic circuit (EC → CA2 → CA1), directly linking EC input to hippocampal CA1 output (*Jones and McHugh, 2011*). How do the EC inputs trigger CA2 AP output, given that synaptic responses at distal dendrites are normally severely attenuated by the cable properties of the dendrites? In this study, we report the PP inputs to distal dendrites in CA2 reliably initiate dendritic $Na^+$ spikes that are necessary to trigger axonal AP output in response to a single or a burst of PP stimuli. Furthermore, these spikes can overcome strong inhibition to elicit AP output. In contrast, activation of PP inputs to distal dendrites of CA1 PNs with a single stimulus fail to elicit dendritic spikes or somatic APs. Through computational modeling based on morphological reconstructions of CA2 and CA1 PNs, we find that the distinct dendritic geometry of CA2 PNs contributes to the ability of CA2 neurons to efficiently couple dendritic $Na^+$ spikes to AP output. Thus our data provide a striking example of how dendritic structure and functional properties control the ability of dendritic $Na^+$ spikes to couple synaptic input at distal dendrites to axonal AP initiation. In this manner, CA2 PN dendritic $Na^+$ spikes ensure the efficient propagation of cortical information through the EC → CA2 → CA1 disynaptic pathway.

## Results

### A single PP stimulus is capable of evoking APs in CA2, but not CA1 or CA3, PNs

Our laboratory found that CA2 PNs receive strong excitatory inputs from EC and fire APs with high probability in response to a brief burst of stimuli delivered to the EC PP axons (5 pulses at 100 Hz). In

contrast, the same PP stimuli generate a smaller synaptic response in CA1 PNs that is usually insufficient to elicit spike output (*Chevaleyre and Siegelbaum, 2010*). In this study, we first re-investigated the input–output relation between distal synaptic stimulation strength and sub-threshold EPSP size recorded in the soma of CA2 PNs. In these experiments the stimulation electrode was placed in SLM of the CA1 region as before but was closer to the CA1/CA2 border (~50 µm) than in our previous study (~200 µm). Consistent with our previous findings (*Chevaleyre and Siegelbaum, 2010*), the sub-threshold EPSP amplitude in CA2 PNs evoked by PP stimulation was 5–6 times larger than that observed in CA1 (*Figure 1A–C*). Moreover, the EPSP in CA2 PNs was slightly larger than that seen in our previous study, most likely due to the closer proximity to CA2 of the stimulation electrode.

We previously surmised that CA2 synaptic responses elicited by a focal stimulating electrode placed in SLM of CA1 were generated by activation of EC inputs from LIII pyramidal neurons, because their axons form dense projections throughout the CA1 SLM layer (*Chevaleyre and Siegelbaum, 2010*). However, recent studies indicate that CA2 PNs receive input largely from EC LII neurons (*Cui et al., 2013*; *Hitti and Siegelbaum, 2014*; *Kohara et al., 2014*). We investigated whether our focal stimulating electrode in SLM of CA1 might also activate EC LII axons by recording from dentate gyrus granule cells, which receive exclusive innervation from EC LII (*Witter, 2007*). Indeed, we find that stimulation in SLM of CA1 evokes EPSPs in the granule cells (*Figure 1—figure supplement 1*), consistent with a tracing study that reported the presence of EC LII axons in SLM of CA1 (*Tamamaki and Nojyo, 1993*). Thus we conclude that the EPSPs recorded in CA2 PNs in response to distal stimulation in SLM of CA1 likely results from the activation of inputs from EC LII.

In the vast majority of CA2 PNs tested (35/42 cells), a single electrical stimulus (up to 48 V) delivered to the PP was sufficient to elicit a large, fast AP that was >80 mV in amplitude when recorded in the CA2 soma (*Figure 1D,E*). The cumulative distribution of spike threshold was well fit by a sigmoidal function (*Figure 1E*). A moderate stimulus strength (28 V) was sufficient to elicit spiking in about half of all CA2 PNs studied, with a small fraction (~5%) firing APs with a very weak stimulus (12 V) (*Figure 1E*). By contrast, CA1 PNs never fired a spike in response to a single PP stimulus, even at strengths up to 60 V (with the stimulating electrode located within 50 µm of the CA1 PN as in our CA2 recordings), but required a burst of high-frequency PP stimuli to fire APs (e.g. *Figure 2D*, *Figure 3B*, *Figure 4B*), consistent with a previous report (*Jarsky et al., 2005*).

To rule out the possibility that CA2 spike firing was an artifact of washout caused by whole-cell recording and to measure the collective activity of a population of neurons, we measured the extracellular population spike (PS) in response to PP stimulation using an extracellular field recording electrode placed in stratum pyramidale (SP) of the CA1, CA2, or CA3b region (*Figure 1F*). Consistent with the whole-cell recordings, a prominent PS was elicited by a single stimulus in the CA2 region. Moreover the CA2 PS input–output curve was fit by a sigmoidal function that closely matched that seen with whole-cell recordings (*Figure 1F*). By contrast, we failed to detect a measurable PS in the cell body layer of the CA1 or CA3b region (*Figure 1F*). The electrically evoked APs were eliminated by blockade of fast glutamatergic synaptic transmission, using bath application of NBQX (20 µM) and D-APV (50 µM) (*Figure 1D*, n = 4). Thus, the spikes were driven by synaptic excitation, not by direct electrical stimulation of CA2 dendrites.

The efficient triggering of action potential output from CA2 PNs in response to PP stimulation is surprising given that the largest EPSP observed in the CA2 PN soma was ~11 mV below the voltage threshold of AP output in response to somatic current injection (*Figure 1G–I*, see also *Figure 2C*). Thus, even though the PP EPSP is fivefold larger in CA2 than CA1 PNs, the mechanism coupling synaptic input to AP output is unclear. In the remainder of this study, we address the hypothesis that dendritic spikes are critical for enabling the distal synaptic inputs to trigger this output.

## Single PP stimuli trigger dendritic spikes in CA2, but not CA1, PNs

Initial evidence for the triggering of dendritic spikes came from inspection of CA2 PN voltage responses to PP stimuli that were just sub-threshold for eliciting somatic action potentials. Nearly all CA2 PNs fired APs in response to a PP stimulus that was strong enough to induce a somatic EPSP greater than 15.6 ± 0.6 mV (ranging from 7.7–20.3 mV, n = 27, *Figure 1G–I*). In a subset of CA2 PNs (n = 9/42 cells), PP stimuli that were just below threshold for eliciting an AP evoked a somatic voltage response with a rapidly rising phase of depolarization not seen in voltage responses evoked by weaker PP stimuli (*Figure 2—figure supplement 1A*). In cells that displayed this rapid voltage response, a small increase in stimulus strength usually evoked a full-blown AP (*Figure 2—figure supplement 1A*).

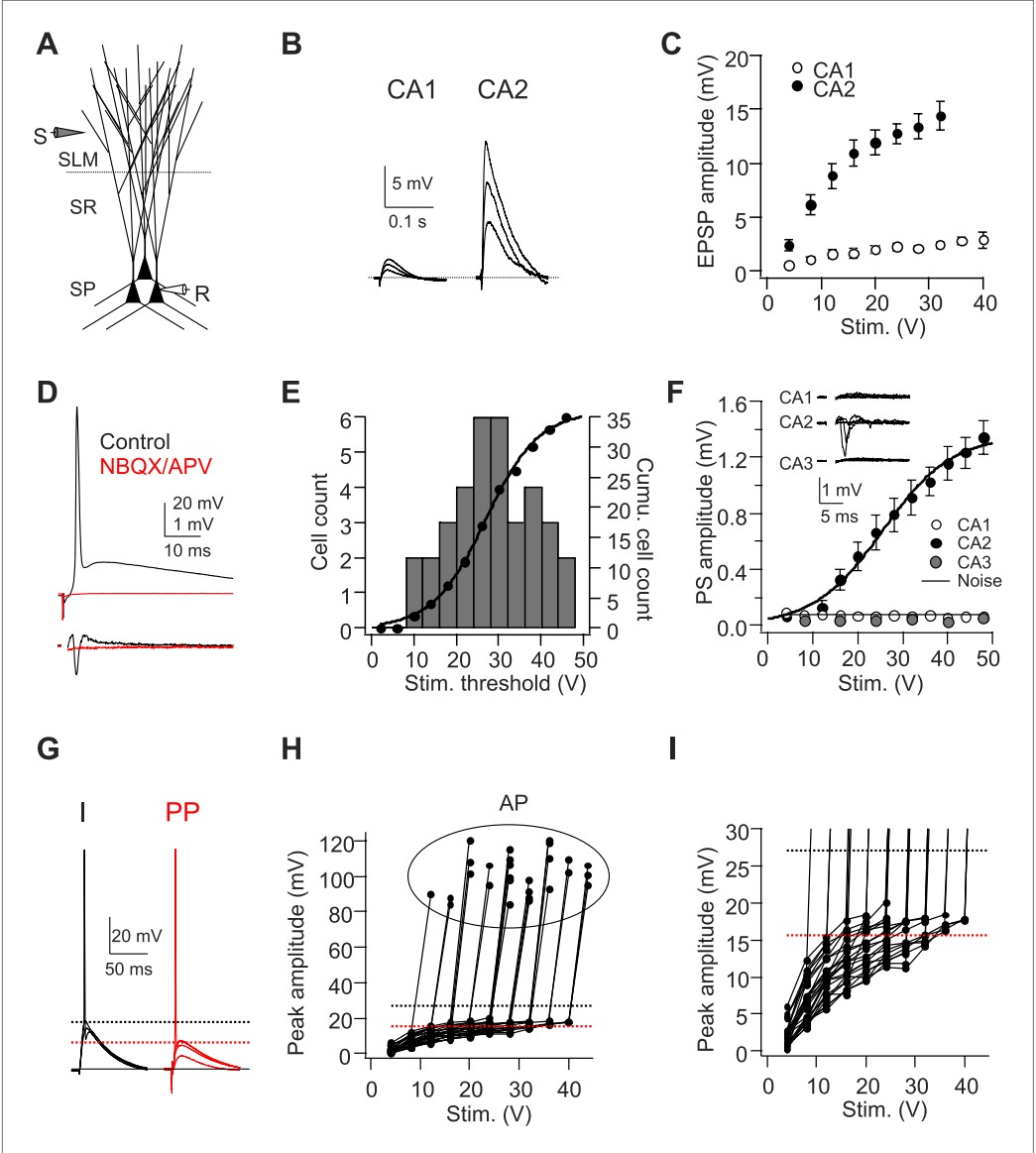

**Figure 1**. A single stimulus delivered to the perforant path (PP) evokes APs in CA2 PNs. (**A**) Diagram illustrating the configuration for the experiment. SLM: stratum lacunosum-moleculare, SR: stratum radiatum, SP: stratum pyramidale. S: stimulating electrode, R: recording electrode. (**B**) Sample traces of EPSPs evoked by PP stimuli using an electrode placed in SLM of CA1. (**C**) Mean input–output curves of somatic EPSPs in CA1 (n = 5) and CA2 (n = 6-12) PNs. (**D**) Simultaneous whole-cell recording from a CA2 PN (top) and extracellular field potential recording from the CA2 cell body layer (bottom) in the absence (black traces) or the presence (red traces) of 20-μM NBQX and 50-μM D-APV. Top: somatic AP evoked by a single PP stimulus. Bottom: extracellular population spike (PS) in CA2 cell body layer. (**E**) Histogram (bars) and cumulative plot (circles) of PP stimulus threshold required to evoke APs in different CA2 PNs (n = 42 cells). (**F**) Mean input–output curves of PS in CA1 (n = 5), CA2 (n = 12), and CA3b (n = 6) cell body layers evoked by single PP stimuli. Inset: sample traces of PS in response to a single PP stimulus recorded in CA1, CA2, and CA3b cell body layers. The noise level was measured using a section of baseline that did not exhibit a PS. (**G**) Sample traces of sub-threshold and AP waveforms in response to somatic current injection (I) and PP stimulation (PP). The dashed lines indicate mean minimal somatic depolarization required to fire APs in response to somatic current injection (black) and PP stimulation (red), respectively. (**H**) The pooled data from CA2 PNs which fire APs in response to PP stimulation (n = 27). Peak somatic voltage amplitude plotted against stimulating intensity. (**I**) Expanded view of sub-threshold EPSP response in (**H**). Note, the dashed lines in (**H**) and (**I**) indicate mean minimal somatic depolarization required to fire APs in response to somatic current injection (black, 27.2 ± 1.2 mV, n = 5) and PP stimulation (red, 15.6 ± 0.6 mV, n = 27), respectively.

*Figure 1. Continued on next page*

*Figure 1. Continued*

The following figure supplement is available for figure 1:

**Figure supplement 1**. Axons from EC layer II project to CA1 SLM.

This rapidly rising depolarization is similar to the somatic spikelets associated with dendritic Na⁺ spikes reported in other studies (*Losonczy and Magee, 2006*; *Losonczy et al., 2008*; *Remy et al., 2009*; *Muller et al., 2012*).

To further characterize the components of the voltage response, we analyzed their maximal rate-of-rise (dV/dt) (*Figure 2—figure supplement 1B,C*). Although dV/dt initially increased linearly with peak EPSP amplitude, as the EPSP reached values around 15–20 mV, dV/dt increased sharply in a non-linear manner. This suggests that large EPSPs are sufficient to trigger a non-linear membrane response that likely reflects the firing of a dendritic spike (*Figure 2—figure supplement 1B,C*). Consistent with this view, the 20–80% rise time of the EPSP also decreased non-linearly as the peak EPSP amplitude reached values above ~15 mV (*Figure 2—figure supplement 1D*).

As the rapidly rising spikelets were observed in only a subset of our recordings, we wondered whether they were a consistent feature of CA2 responses to PP stimuli but were normally masked by the much larger somatic AP (e.g. *Figure 1D,G*, *Figure 2A*). In agreement with this idea, a phase-plane plot of dV/dt vs membrane voltage showed that APs induced by PP stimulation in CA2 PNs consistently exhibited an initial rapid dV/dt signal at potentials sub-threshold to the main spike (*Figure 2A,B*). Such an early rapid phase of depolarization was not observed when APs were elicited by somatic current injection, suggesting that they do indeed represent dendritic spikes (*Figure 2A,B*). Furthermore, we compared the threshold to fire APs induced by PP stimulation with that induced by somatic current injection. The AP threshold (defined as the somatic voltage at which dV/dt reached 50 V/s) was significantly lower in response to PP stimulation (−51.6 ± 1.0 mV) compared to somatic current injection (−44.3 ± 0.8 mV, n = 17, p < 0.001; *Figure 2C*), indicating that dendritic spikes may indeed be required for PP input to trigger somatic APs.

We next asked whether dendritic spikes were also a feature of PP-evoked action potentials in CA1 PNs. As a single PP stimulus was ineffective in triggering a somatic action potential in CA1 PNs, we used a 50-Hz burst of 5 strong PP stimuli to evoke somatic spiking. In marked contrast with our results with CA2 PNs, neither our somatic voltage recordings nor phase-plane plots showed evidence of a spikelet (*Figure 2D,E*). Moreover, individual spikes, phase-plane plots, and voltage threshold of somatic APs induced by PP stimulation were identical with those induced by somatic current injections (*Figure 2D–F*). These results suggest that in CA1 PNs, somatic depolarization resulting from temporal summation of PP-evoked EPSPs, rather than dendritic spikes, drives AP output.

To examine more directly whether dendritic spikes consistently underlie the CA2 PN somatic AP, we prevented AP firing by injecting negative current to hyperpolarize the somatic membrane to −82.5 ± 2.8 mV (n = 10) (*Figure 3A*). Strikingly, this revealed that PP stimulation consistently evoked a rapidly rising spikelets (dV/dt of 20.6 ± 2.6 V/s; n = 10) in response to PP stimulation (*Figure 3A*). When we applied repeated single PP stimuli of a constant strength near the threshold for eliciting somatic APs, the CA2 PN membrane response often fluctuated between a full-blown AP response and a spikelet (*Figure 3A*). Overlaying these responses showed that the spikelets preceded the full-blown APs (*Figure 3A*), suggesting that the dendritic spikes triggered the full-blown APs. This was particularly evident in plots of dV/dt, where the spikelet waveform observed in isolation could be identified in traces associated with full-blown APs immediately prior to the AP response (*Figure 3A*). In marked contrast, we failed to observe spikelets in CA1 soma upon membrane hyperpolarization (*Figure 3B*, n = 9).

As a second means of recording spikelets in the absence of somatic APs, we locally applied tetrodotoxin (TTX, 1 μM) to the soma of CA1 or CA2 PNs while maintaining the neurons at their initial resting potential (*Figure 4*). In CA2 PNs, this manipulation revealed the consistent presence of spikelets (dV/dt = 36.4 ± 3.0 V/s, n = 8) in response to PP stimuli (*Figure 4C–I*). These spikelets provide a substantial somatic depolarization (33.6 ± 2.8 mV, n = 8) that would normally trigger a somatic action potential in the absence of TTX. Overlaying dV/dt in phase-plane plots in the absence and the presence of TTX again showed that the spikelets preceded the full-blown APs (*Figure 4E*). In contrast, we failed to observe spikelets with TTX applied to the soma of CA1 PNs (*Figure 4A,B*, n = 4). Taken

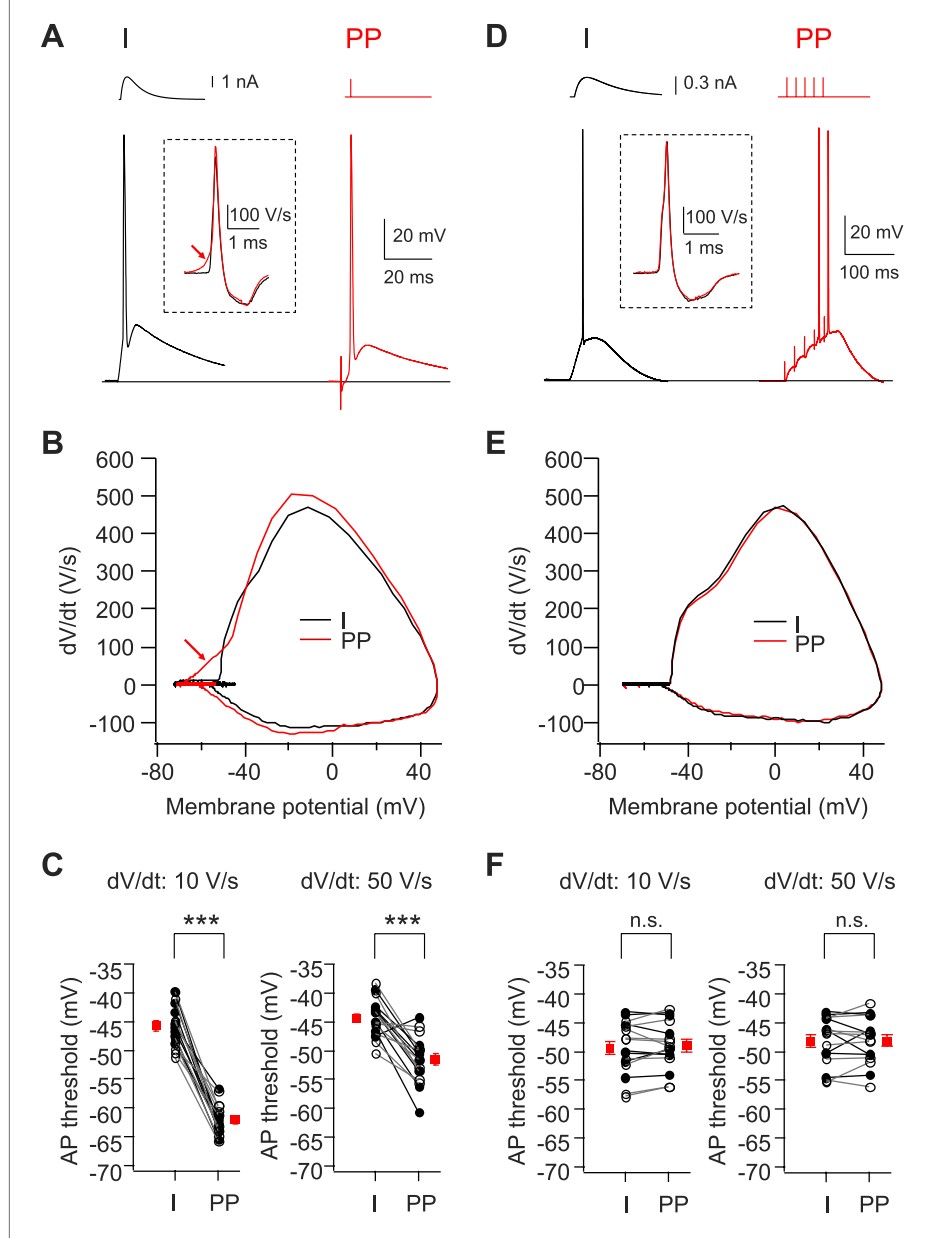

**Figure 2**. Voltage threshold of APs evoked by PP stimulation in CA2, but not CA1, PNs is lower than threshold of APs evoked by somatic current injection. (**A**) Sample traces of somatic voltage responses in response to somatic current injection (I) and a single PP stimulus (PP) from a CA2 PN. Inset shows the dV/dt of the AP waveforms. Arrow indicates the dendrite spike with PP stimulation. (**B**) Phase-plane plots of dV/dt vs instantaneous voltage from data shown in (**A**). Note, the arrow indicates a dendrite spike preceding a full-blown AP. (**C**) Pooled data of AP threshold induced by somatic current injections (I) vs PP stimulation (PP) (n = 17). Filled circle: constant current injection. Open circle: EPSC-like current injection. AP threshold defined as the somatic voltage at which dV/dt exceeds 10 V/s (left) or 50 V/s (right). ***p < 0.001. (**D**) Sample traces of somatic voltage responses in response to somatic current injection (I) and high-frequency (50 Hz, 5 pulses) burst PP stimulation (PP) from a CA1 PN. Inset shows the dV/dt of AP waveforms. (**E**) Phase-plane plots of dV/dt vs instantaneous voltage from data shown in (**D**). Note, the phase plot from somatic current injection (I) is identical with that from PP stimulation (PP). (**F**) Pooled data of AP threshold induced by somatic current injections (I) vs PP stimulation (PP) (n = 16). Filled circle: constant current injection. Open circle: EPSC-like current injection. AP threshold defined as the somatic voltage at which dV/dt exceeds 10 V/s (left) or 50 V/s (right). n.s., not significant.

The following figure supplement is available for figure 2:

**Figure supplement 1**. A single stimulus to the PP triggers dendritic spikes (D spikes) in CA2 PNs.

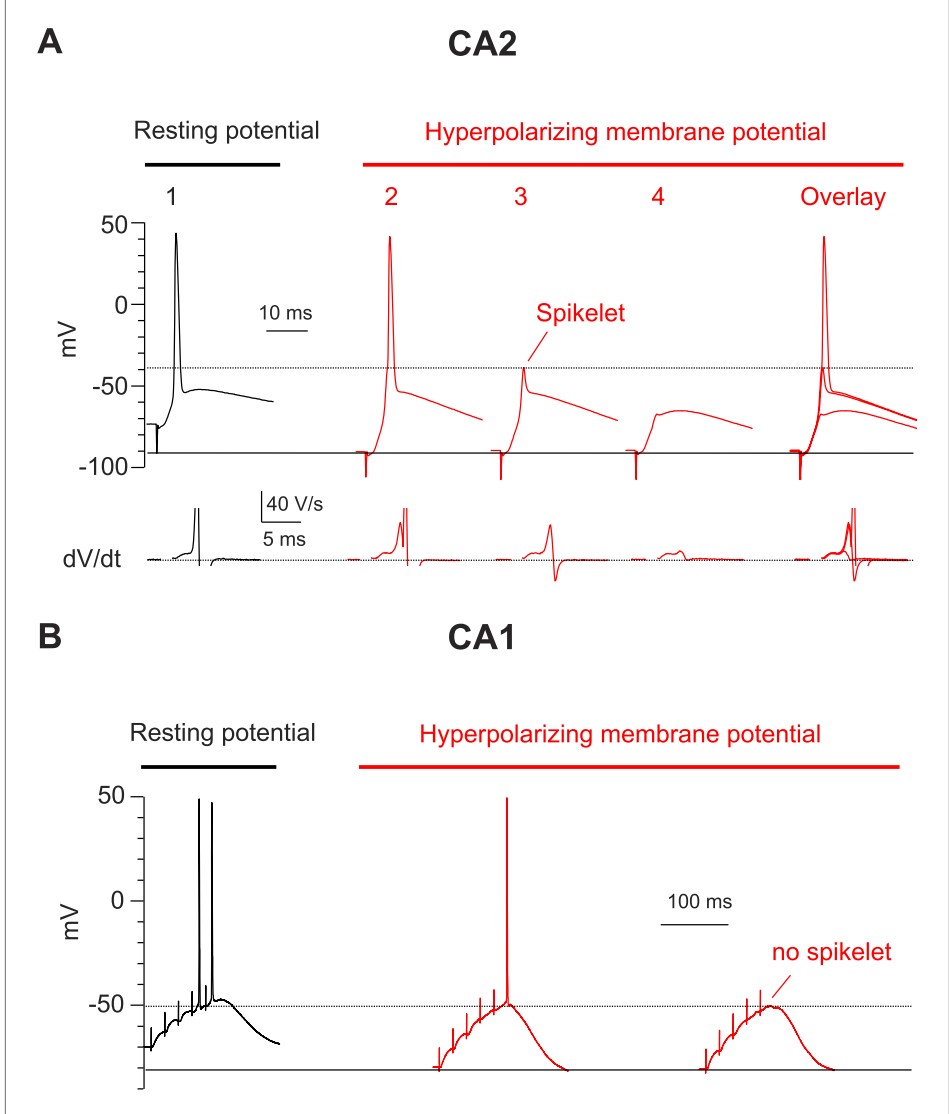

**Figure 3**. Hyperpolarization of membrane potential reveals prominent spikelets at the soma of CA2, but not CA1, PNs in response to PP stimulation. (**A**) Top: a PP stimulus in a CA2 PN reliably elicited a somatic AP at the normal resting potential (trace 1). Hyperpolarization of the resting membrane (traces 2–4) reveals that a PP stimulus with constant strength induced spikelets that variably succeeded (trace 2) or failed (trace 3) in triggering a somatic AP from the hyperpolarized potential. Trace 4 shows a very weak spikelet response to the PP stimulus. Bottom: dV/dt for corresponding voltage responses on top. dV/dt of APs is truncated. Right: overlay of traces 2–4. (**B**) High-frequency burst PP stimulation (50 Hz, 5 pulses) in a CA1 PN triggers somatic APs at the resting potential (left). Hyperpolarization of the resting membrane fails to reveal spikelets (right).

together, we conclude that single PP stimuli are sufficient to evoke prominent dendritic spikes, which result in high-amplitude somatic spikelets in CA2 but not CA1 PNs.

## PP-evoked dendritic spikes are necessary to generate AP output in CA2 PNs in response to PP stimulation

Next, we asked whether PP-driven dendritic spikes are necessary to generate AP output in CA2 PNs. As discussed above, one indication for the necessity of dendritic spikes is our finding that the threshold to fire a somatic AP with PP stimulation is negative to the threshold with somatic current injection (*Figure 5—figure supplement 1*). To directly determine whether dendritic spikes are required for eliciting a CA2 PN somatic action potential under the conditions of our experiments, we applied TTX

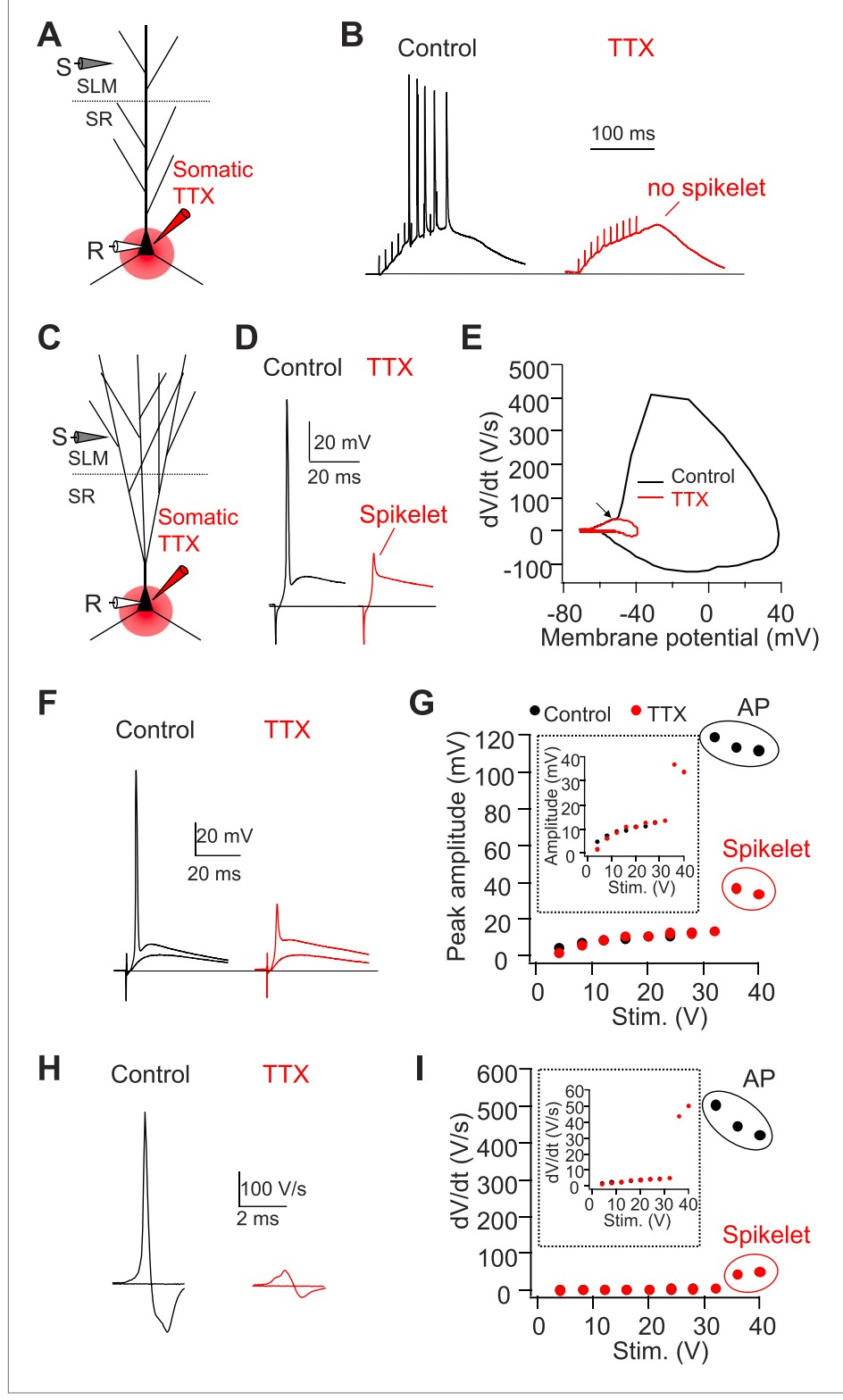

**Figure 4**. Somatic TTX application reveals prominent spikelets at the soma of CA2, but not CA1, PNs in response to PP stimulation. (**A**) Diagram illustrating the configuration of CA1 PN experiment as shown in (**B**). (**B**) Sample traces of somatic voltage response in a CA1 PN to high-frequency burst stimulation (100 Hz, 10 pulses) in the absence (control) or the presence of somatic TTX (somatic TTX). Note the absence of spikelets at the soma.
*Figure 4. Continued on next page*

*Figure 4. Continued*

(**C**) Diagram illustrating the configuration of CA2 PN experiment shown in (**D–I**). (**D**) Sample traces of somatic voltage response in a CA2 PN to a single PP stimulus in the absence (control) or the presence of somatic TTX (TTX). Note the presence of a prominent spikelet during somatic TTX application at the soma of CA2 PN. (**E**) A phase-plane plot from the traces shown in (**D**). Note the overlap of the initial rising phase in control and TTX (arrow). (**F**) Superimposed traces of somatic voltage response of a CA2 PN to suprathreshold and subthreshold PP stimuli in the absence or the presence of somatic TTX. (**G**) Input–output of somatic voltage response from the CA2 PN shown in (**F**). Inset shows an expanded plot of the sub-threshold somatic voltage response. Note, the spikelet amplitude reaches >35 mV, providing ~20 mV extra somatic depolarization on top of the EPSP. (**H**) Sample traces of dV/dt from (**F**). (**I**) Input–output curve of dV/dt from the CA2 PN shown in (**F–H**). Inset shows an expanded plot of sub-threshold dV/dt. Note, dV/dt of the spikelets reaches ~50 V/s.

locally to the proximal dendrites of CA2 PNs, which should block dendritic Na$^+$ spikes. Indeed, this manipulation fully blocked the ability of single PP stimuli to elicit somatic APs (*Figure 5A–D*, n = 6). After blockade of dendritic Na$^+$ spikes, the EPSP evoked by strong PP stimulation maximally depolarized the soma by ~20 mV positive to the resting potential (−70 to −73 mV) (*Figure 5D*), below the threshold for driving AP output by somatic current injection. Importantly, the local TTX application exerted a selective effect on dendritic excitability and did not alter PP synaptic transmission or somatic excitability. Thus, there was no change in the sub-threshold PP EPSP or AP firing in response to somatic current injection (*Figure 5E–H*). These results strongly suggest that dendritic spikes in CA2 PNs are mediated by TTX-sensitive voltage-gated Na$^+$ channels (see below for additional evidence) and are necessary for somatic AP initiation with a single PP stimulus.

In awake-behaving animals, EC neurons often fire in high-frequency bursts rather than isolated single APs (*Burgalossi et al., 2011*). It is thus possible that temporal summation of somatic depolarization driven by EC bursting may be sufficient to elicit AP output in CA2 PNs without the need for dendritic spikes. However, a phase-plane plot revealed the presence of an early rapid rising voltage response preceding the full-blown somatic spikes in response to a 50 Hz burst of 5 PP stimuli (*Figure 6A,B*). Moreover, application of somatic TTX revealed the consistent presence of somatic spikelets (*Figure 6C,D*, n = 6) that precede AP initiation in the absence of TTX (*Figure 6D*). Importantly, local application of TTX to the proximal dendrites fully blocked the ability of PP burst stimuli to drive AP output (*Figure 6E,F*, n = 3). Thus, under the conditions of our experiments, cortically-driven dendritic Na$^+$ spikes were necessary for AP output in CA2 PNs with either single or bursts of PP input.

## Latency to fire APs in response to PP burst stimulation is shorter in CA2 than CA1

Dendritic Na$^+$ spikes can influence the timing of AP output and thus have been proposed to contribute to temporal coding (*Ariav et al., 2003*). To address whether dendritic Na$^+$ spikes affect the timing of AP output, we measured population spikes in the CA1 and CA2 stratum pyramidale (SP) cell body layers in response to a burst of PP stimuli (5 pulses at 50 Hz). In CA1, a measurable PS was never observed following the first PP stimulus and was only detected during the later stimuli in a burst, resulting in a latency to firing of 20 ms or more (*Figure 6—figure supplement 1*). In contrast, the PS amplitude in CA2 was largest in response to the first stimulus of a burst and gradually diminished with successive stimuli (*Figure 6—figure supplement 1*). The decrease in PS response was not due to synaptic depression because the paired-pulse ratio (PPR) of PP synapses in CA2 PNs exhibited facilitation rather than depression (PPR = 1.39 ± 0.06, n = 17). Thus, CA2 PNs fired precisely and immediately in response to PP stimulation (latency of the first PS = 4.4 ± 0.1 ms, n = 5), suggesting that dendritic Na$^+$ spikes enable CA2 PNs to respond rapidly to EC burst firing with a high degree of temporal fidelity.

## CA2 dendrites generate Na$^+$ spikes in response to PP stimulation

To examine dendritic spike firing in response to PP stimulation more directly, we measured the local field responses simultaneously from two extracellular recording electrodes placed near the middle of the axial axis of the apical dendrites in the stratum radiatum layer (SR) and in the SP layer of CA1, CA2, or CA3 regions (*Figure 7A*). CA2 dendrites generated an active excitatory current response to PP stimulation, manifested as a negative field voltage response in recordings from SR (*Figure 7A,B*). In contrast, PP stimuli never evoked an active response in SR of CA1 or CA3. As the excitatory synaptic response is local

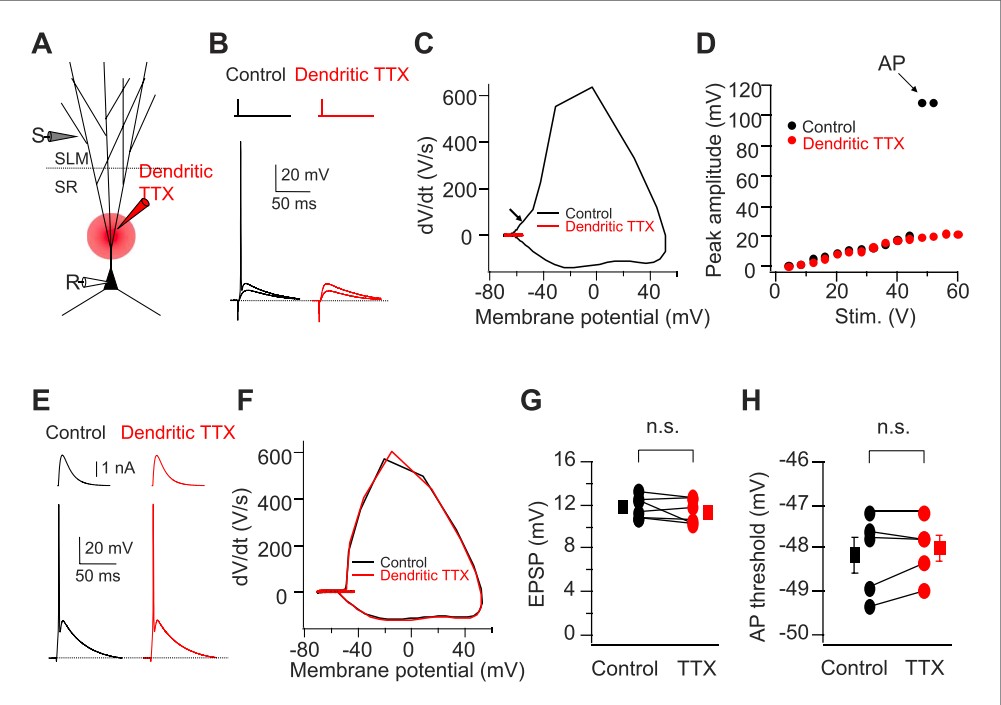

**Figure 5**. Cortically driven dendritic Na⁺ spikes are necessary to fire APs in CA2 PNs in response to a single PP stimulus. (**A**) Diagram illustrating the configuration of the experiment. (**B**) Superimposed voltage responses of CA2 PN in response to suprathreshold and subthreshold PP stimuli. Responses obtained in the absence (Control) or the presence of TTX applied to dendrites in SR (Dendritic TTX). Note dendritic TTX blocks AP in response to strong stimulus but does not alter subthreshold EPSP in response to weaker stimulus. (**C**) Phase-plane plots of dV/dt vs instantaneous voltage from data shown in (**B**). Arrow indicates a dendritic spike preceding a full-blown AP. (**D**) Input–output curve of somatic voltage response of an individual CA2 PN to PP stimulation in the absence of TTX (Control) or during local TTX application in SR (Dendritic TTX). (**E**) Sample traces of AP waveforms in response to somatic EPSC-like current injection. Responses obtained in the absence (Control) or the presence of TTX applied to dendrites in SR (Dendritic TTX). (**F**) Phase-plane plots of dV/dt vs instantaneous voltage from data shown in (**E**). Note lack of the rising phase preceding a full-blown AP seen in (**C**). (**G** and **H**) Sub-threshold EPSP evoked by PP stimulation (n = 6) and AP voltage threshold evoked by somatic EPSC-like current injection (n = 5) in the absence (Control) and the presence of TTX applied to dendrites in SR (TTX) for individual experiments (circles) and mean (squares). Error bars show SEM. SEM was smaller than symbol. n.s., not significant.

The following figure supplement is available for figure 5:

**Figure supplement 1**. Dendritic Na⁺ spikes are necessary to fire APs in CA2 PNs.

to the site of PP input in SLM, the negative field response in SR must be generated by voltage-gated excitatory conductances. Importantly, the amplitude of the active response in SR was correlated with PS size in SP (somatic spike; R = 0.91, *Figure 7C*). Furthermore, the SR potential preceded the PS in SP (*Figure 7D*), suggesting that dendritic Na⁺ spikes precede somatic APs. Moreover, the difference in latency between SR and SP active responses was correlated with the distance between the dendritic and somatic recording sites (*Figure 7D*). These results suggest that the dendrites of CA2 PNs, but not CA1 or CA3 PNs, fire spikes that propagate to the soma of CA2 PNs to trigger AP output.

To characterize further the ionic mechanisms of the dendritic spikes and the active dendritic properties of CA2 PNs, we recorded voltage responses directly from CA2 dendrites under whole-cell current clamp conditions. CA2 dendrites had distinct electrophysiological properties from CA1 dendrites, including lack of voltage sag, a late depolarizing response to hyperpolarizing current injection characteristic of $I_h$, and lack of AP adaptation (CA2 recordings of *Figure 8B* compared to CA1 recordings in *Figure 8—figure supplement 1*). The identity of all recorded CA2 PNs was subsequently verified by biocytin staining based on morphology, including a lack of large thorny excrescences (a characteristic

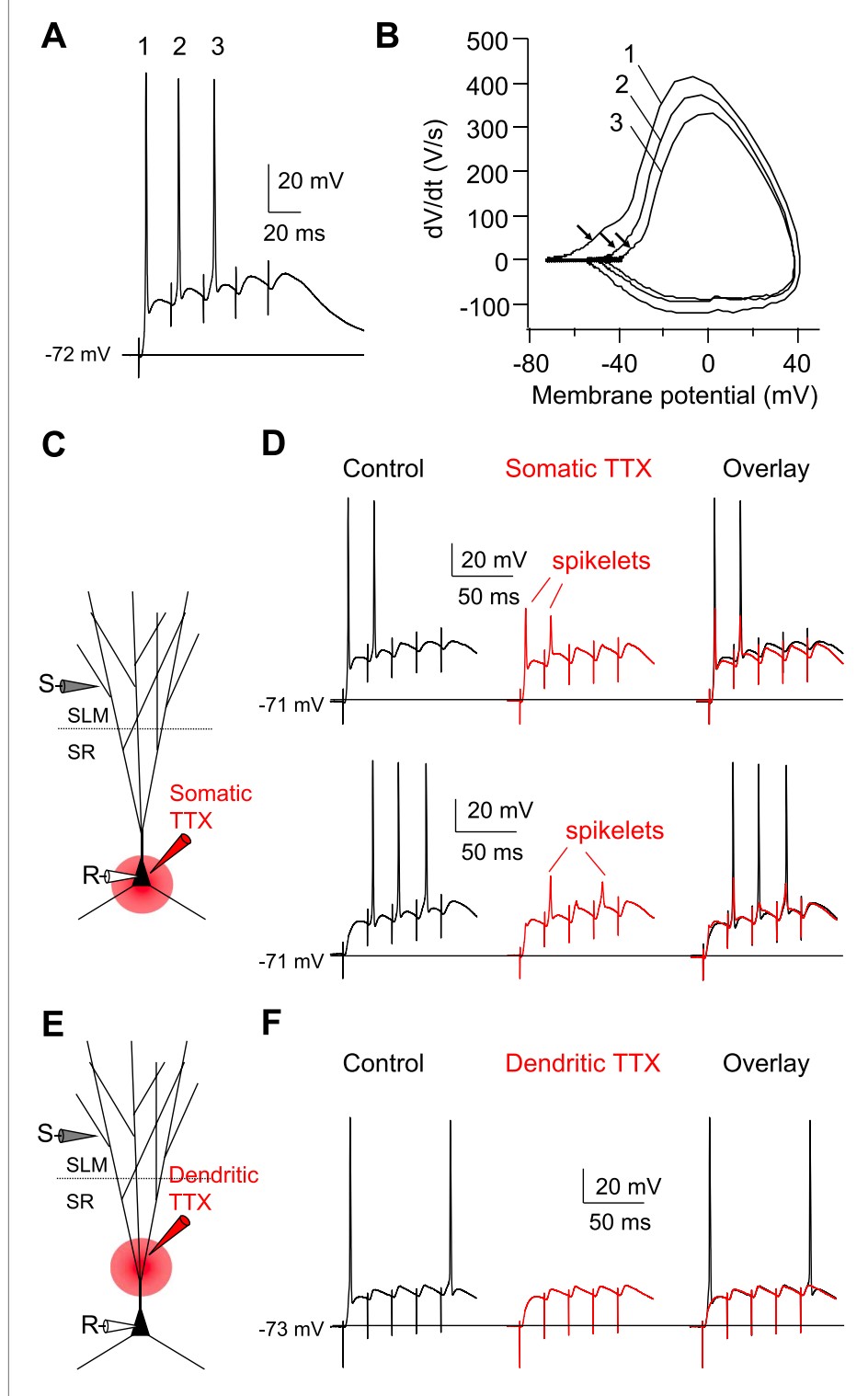

**Figure 6**. Dendritic Na[+] spikes evoked by high-frequency burst stimulation to PP are necessary to fire APs in CA2 PNs. (**A**) Sample somatic voltage response to high-frequency PP burst stimulation (50 Hz, 5 pulses). (**B**) Phase-plane plot from the traces shown in (**A**). The numbers (1–3) correspond to AP waveforms shown in (**A**). Note, the arrows indicate the rapid rising phase (dendritic Na[+] spikes) preceding the full-blown APs. (**C**) Diagram illustrating the configuration of the experiment shown in (**D**). (**D**) Sample traces of somatic spikelets revealed by somatic TTX application in response to 48 V (top) or 32 V (bottom) high-frequency PP burst stimulation (50 Hz, 5 pulses).
*Figure 6. Continued on next page*

*Figure 6. Continued*

Left: control, middle: somatic TTX, right: overlay. (**E**) Diagram illustrating the configuration of the experiment shown in (**F**). (**F**) Sample traces of somatic voltage response to high-frequency PP burst stimulation (50 Hz, 5 pulses). Left: control, middle: dendritic TTX, right: overlay. Note, APs are blocked by dendritic TTX application.

The following figure supplement is available for figure 6:

**Figure supplement 1**. Latency to fire APs in response to PP burst stimulation is shorter in CA2 than CA1.

of CA3 PNs), the presence of a large cell body, and distinct dendritic branching patterns (*Figure 8A*; *Ishizuka et al., 1995*). We found that dendritic spikes could be reliably triggered by brief depolarizing current pulses in all CA2 dendrites examined, with a current threshold of 1.7 ± 0.2 nA (n = 10) (*Figure 8C*). Furthermore, the dendritic spikes were generated by voltage-gated $Na^+$ channels as they were blocked by bath application of TTX (0.5 µM, n = 3; *Figure 8C*). Finally, the dendritic recordings directly demonstrated that a single PP synaptic stimulus was able to evoke a dendritic spike (*Figure 8D,E*).

## NMDAR activation is not required for dendritic $Na^+$ spikes in CA2 PNs

In addition to $Na^+$ spikes, apical tuft dendrites can generate local NMDA spikes to enhance synaptic input (*Schiller et al., 2000*; *Nevian et al., 2007*; *Larkum et al., 2009*; *Lavzin et al., 2012*). However, we found that bath applied D-APV (50 µM) did not block dendritic spikes elicited by a single PP stimulus (*Figure 8—figure supplement 2*) and failed to inhibit the PS in the CA2 PN layer elicited with strong current stimulating pulses (>24 V; *Figure 8—figure supplement 2*). However, D-APV did produce a small decrease (20%) in EPSP amplitude (*Figure 8—figure supplement 2*). Thus NMDAR activation is not required for dendritic spike initiation or AP output in response to PP activation in CA2 PNs.

## Dendritic $Na^+$ spikes in CA2 PNs overcome inhibition to trigger APs

The experiments so far described were performed in the presence of GABA receptor antagonists (see 'Materials and methods') to facilitate dendritic excitation. However, under physiological conditions, the distal dendrites of CA2 PNs receive powerful inhibition that can significantly influence their excitation by the EC inputs. We thus asked whether dendritic $Na^+$ spikes can enable CA2 PNs to overcome inhibition and allow the PP inputs to trigger AP output (*Muller et al., 2012*).

We first compared the input–output relation for the CA2 PN somatic postsynaptic potential (PSP) amplitude as a function of PP stimulus strength in the presence or the absence of GABAR antagonists. For stimulus strengths above 12 V, the PP PSP was significantly enhanced by the GABAR antagonists, indicating that PP stimulation did indeed elicit strong inhibition (*Figure 9A*). At an intermediate stimulus strength of 24 V, the PP-evoked PSP depolarized the membrane by only 6.6 ± 0.9 mV with inhibition intact, whereas the PSP size increased to 11.7 ± 1.3 mV when inhibition was blocked (p < 0.001, n = 10). Even at maximum strength (52 V), the PSP amplitude only reached 8.4 ± 1.1 mV with inhibition intact (ranging from 4.3 mV to 13.6 mV, n = 9, excluding responses with dendritic $Na^+$ spikes or APs). Given that the resting potential of CA2 PNs was −75.6 ± 0.7 mV (n = 23) and the AP threshold with somatic current injection was −44.3 ± 0.8 mV (n = 17), this level of depolarization is very far from threshold for eliciting somatic APs. Surprisingly, however, extracellular field recordings showed that a single PP stimulus was, in fact, able to evoke a PS in the CA2 cell body layer even with inhibition present (*Figure 9B*). As expected, the stimulation threshold for eliciting a PS was higher and the PS amplitude was reduced when inhibition was intact compared to when inhibition was blocked (*Figure 9B*).

Somatic whole-cell recordings confirmed that a single PP stimulus was able to elicit AP firing when inhibition was present, although with a reduced probability compared to when inhibition was blocked (*Figure 9C,D*). With inhibition intact, CA2 PNs fired APs following somatic PSPs whose peak amplitude, on average, reached a threshold of 10.99 ± 0.59 mV (*Figure 9E,F*, ranging from 8.6–13.6 mV, n = 6).

In the presence of inhibition, repeated trials of PP stimulation using a constant stimulus strength near threshold elicited variable responses, with full-blown APs, spikelets (dendritic $Na^+$ spikes), or subthreshold PSPs (*Figure 9G*, *Figure 9—figure supplement 1*, n = 8). A close inspection of traces with APs or dendritic spikes and an analysis of dV/dt demonstrated that dendritic $Na^+$ spikes consistently preceded full-blown APs (*Figure 9G*, *Figure 9—figure supplement 1*). Thus, dendritic $Na^+$ spikes

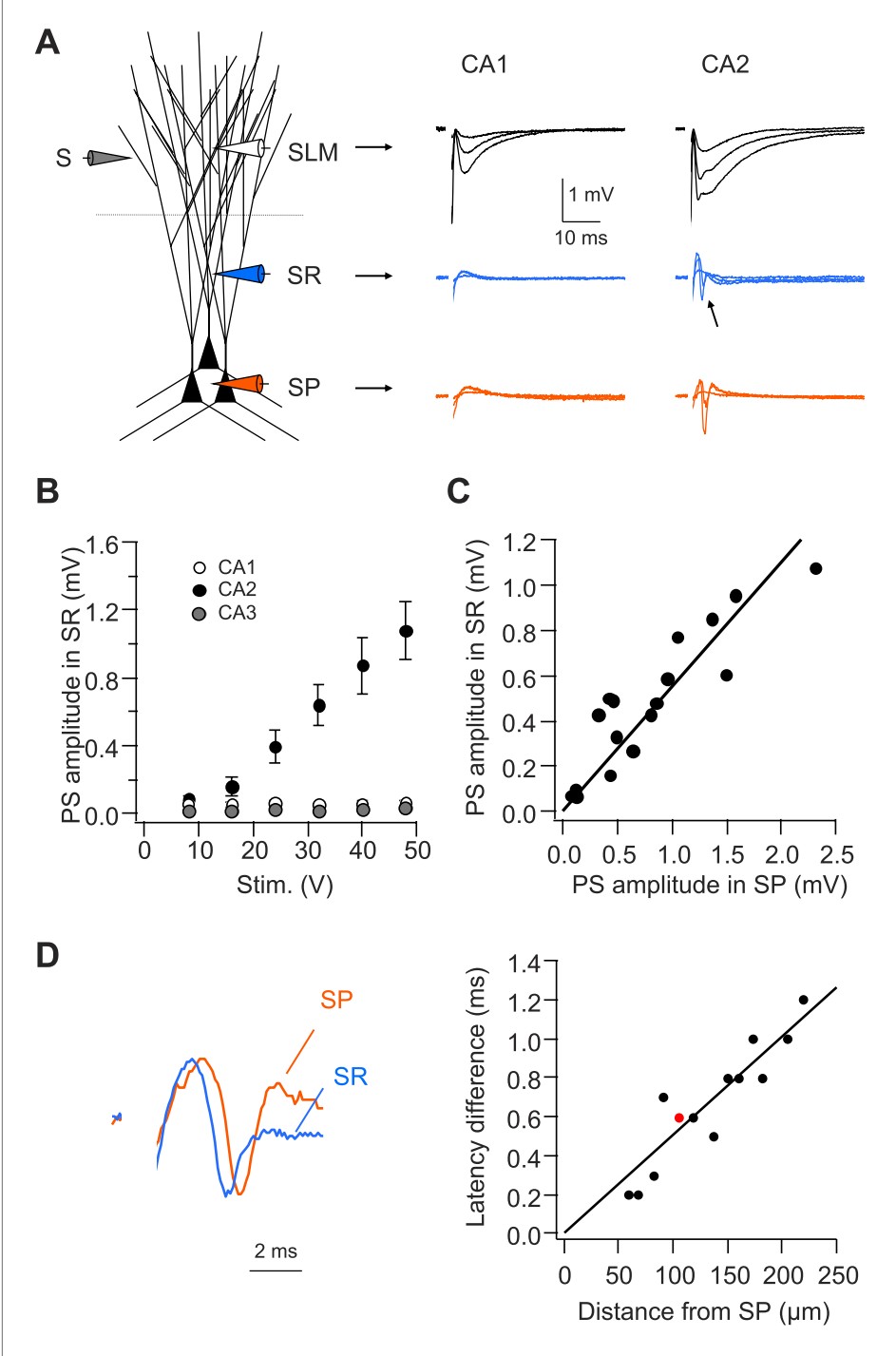

**Figure 7**. CA2, but not CA1 or CA3, dendrites are active in response to PP stimulation. (**A**) Left: diagram illustrating the configuration for extracellular field recording. Right: sample traces of field EPSP (fEPSP) responses in SLM, SR, and SP of CA1 and CA2 regions. Arrow indicates active dendritic response (negative field potential) in SR of CA2. (**B**) Mean input–output curves of PS amplitude in SR of CA1 (n = 4), CA2 (n = 7), and CA3 (n = 6) regions. (**C**) PS amplitude in SR plotted against PS amplitude in SP from the simultaneous field recordings in SR and SP in CA2 (correlation coefficient = 0.91, p < 0.001, n = 17). (**D**) Left: scaled simultaneously recorded field responses in SR and SP in CA2 (red circle in the right panel; distance from SP = 105 µm). Right: the time difference of the response latency between SR and SP plotted against the distance between the two recording electrodes (correlation coefficient = 0.92, p < 0.001, n = 13).

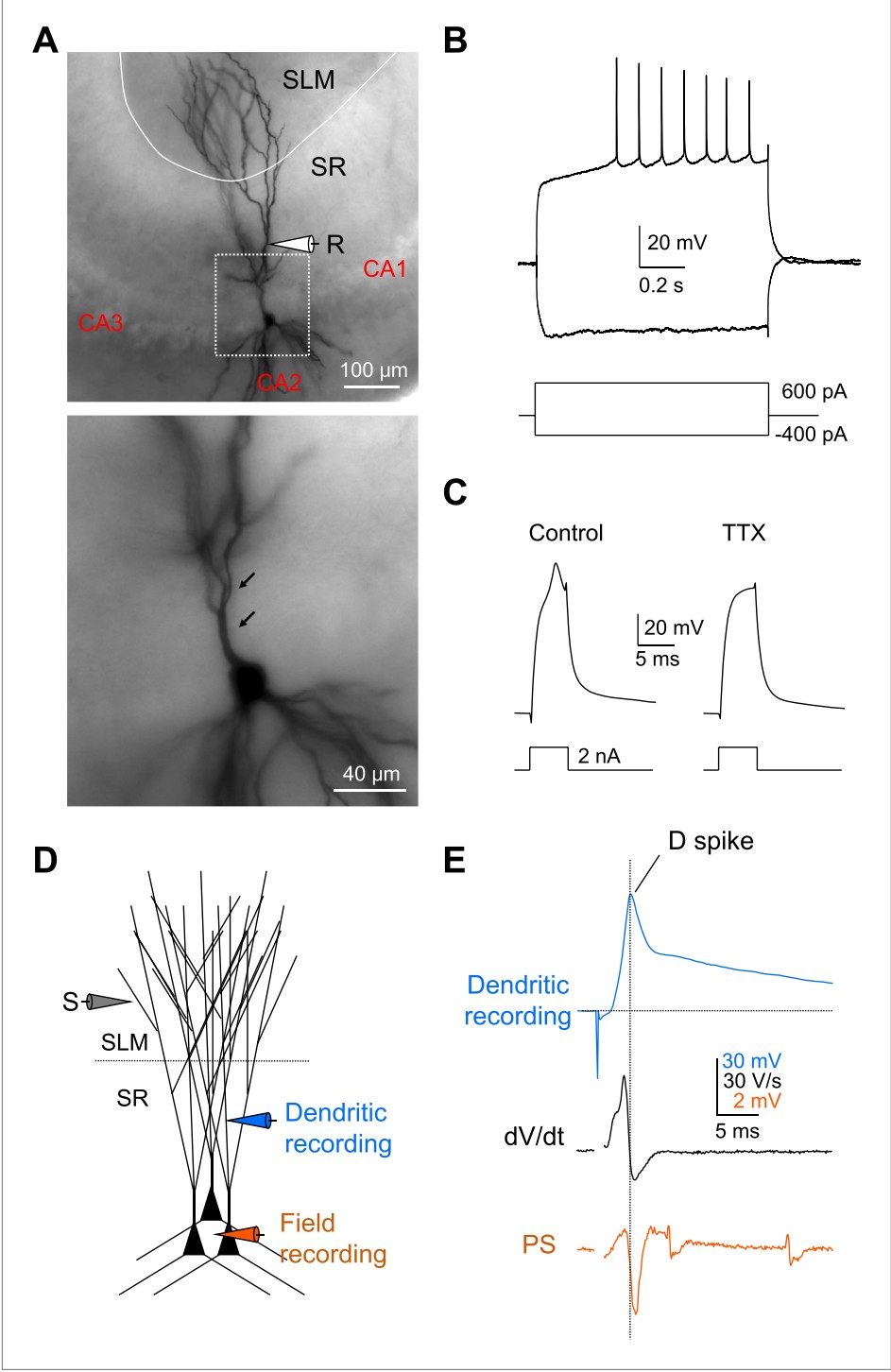

**Figure 8**. Local dendritic Na+ spikes observed with dendritic whole-cell recordings in CA2 PNs. (**A**) Top: a typical CA2 PN filled with biocytin following dendritic whole-cell recording (recording distance: ~125 µm from soma). Bottom: an expanded view of the box shown on top. Arrows show the lack of thorny excrescences, the postsynaptic spines of mossy fiber synapses seen in CA3 PNs. (**B**) The voltage response of a CA2 PN dendrite to local current injection (same neuron as in **A**). (**C**) A dendritic spike (recording distance: ~150 µm) evoked by a 5-ms current pulse in the absence or the presence of TTX (0.5 µM). (**D**) Diagram illustrating the configuration for simultaneous dendritic whole-cell recording and extracellular field recording in CA2 cell body layer shown in (**E**). (**E**) Dendritic

*Figure 8. Continued on next page*

*Figure 8. Continued*

spike (top), dV/dt (middle), and PS (bottom) in response to a single PP stimulus (same neuron as in **C**). Note, dendritic spike precedes PS in CA2 cell body layer.

The following figure supplements are available for figure 8:

**Figure supplement 1**. Dendritic whole-cell recordings in CA1 dendrites.

**Figure supplement 2**. NMDAR activation is not required for dendritic spikes in CA2 PNs.

driven by EC inputs substantially boost somatic depolarization to overcome inhibition and enable CA2 PNs to generate AP output.

## Dendritic architecture contributes to differences in coupling dendritic Na$^+$ spikes to AP output in CA1 vs CA2 PNs

In CA1 PNs, the propagation of dendritic Na$^+$ spikes to the soma is severely attenuated by the dendritic cable properties. As a result, these spikes appear at the soma as small (<5 mV), slowly rising (dV/dt < 10 V/s) spikelets (*Spruston, 2008*). In contrast, the spikelets at CA2 PN soma have a much larger amplitude (25.5 ± 2.6 mV) and rate-of-rise (dV/dt = 36.4 ± 3.0 V/s, n = 8) and so are capable of driving action potential output. Why do CA2 PNs produce such large Na$^+$ spikelets at the soma compared to CA1 PNs?

To explore the factors that contribute to these differences, we constructed morphologically realistic computational models of CA2 and CA1 PNs, based on Neurolucida reconstructions of biocytin-filled cells (*Hines and Carnevale, 1997*). CA2 PN apical dendrites have a number of distinct morphological features compared with CA1 dendrites that might influence spike propagation (e.g. *Figures 8A* and *Figure 10*, *Figure 10—figure supplements 1 and 2*). CA2 neurons extend a single apical dendrite from the soma that, within 50–100 μm, splits into multiple secondary branches that project into SLM. Thus each branch provides an independent direct route for voltage to propagate to the soma. In contrast, CA1 neurons send a single apical dendrite to the border of SR and SLM, where the branch splits into a number of fine secondary and tertiary tuft dendrites.

As there are few quantitative measurements of voltage-gated conductances in the very thin dendrites of mouse CA1 and CA2 PNs, we used the same conductance parameters previously used to model rat CA1 PN dendrites, consisting of a voltage-gated Na$^+$ conductance (G$_{Na}$), delayed rectifier and A-type K$^+$ conductances (G$_{Kdr}$ and G$_{KA}$), and a hyperpolarization-activated cation conductance, $I_h$ (*Jarsky et al., 2005*; see 'Materials and methods').

We lowered the $I_h$ conductance in CA2 PNs relative to that of CA1 PNs to match our experimental measures of voltage sag, a slow depolarizing response that follows the hyperpolarizing response to inward current steps that is characteristic of the activation of $I_h$. We then slightly adjusted the values of G$_{Na}$, G$_{Kdr}$ and G$_{KA}$ so that the excitability of our models matched our experimental results. The final values of these conductances were identical in the CA1 and CA2 models (see 'Materials and methods').

We first asked whether simulated PP synaptic input onto distal dendrites of CA1 and CA2 PNs is capable of generating dendritic spikes that are sufficient to drive AP output. Consistent with our experimental results, strong synaptic stimulation onto CA2 distal dendrites did indeed produce dendritic Na$^+$ spikes that propagated to the soma to trigger AP output. By contrast, in the CA1 PN model, although a similar level of distal synaptic stimulation was able to trigger local Na$^+$ spikes at the apical tufts, these spikes propagated poorly to the soma and failed to initiate AP output (*Figure 10A–D*).

Why do the CA2 dendrites propagate dendritic spikes to the soma more efficiently than the CA1 PN dendrites? One important clue comes from the presence of a large local spike at the main branch point of the apical dendrites. Moreover this spike precedes the somatic AP in response to PP stimulation (*Figure 10D*, *Figure 10—figure supplement 2*). Such findings suggest that the main branch point in CA2 PNs acts as a 'hot spot' that generates large dendritic spikes that trigger AP output. Similar results were obtained in models from a second set of reconstructed CA1 and CA2 PNs (*Figure 10—figure supplement 2*).

To explore further the influence of dendritic architecture on spike firing, we examined the influence of activating an increasing number of secondary or tertiary apical branches in the CA2 model.

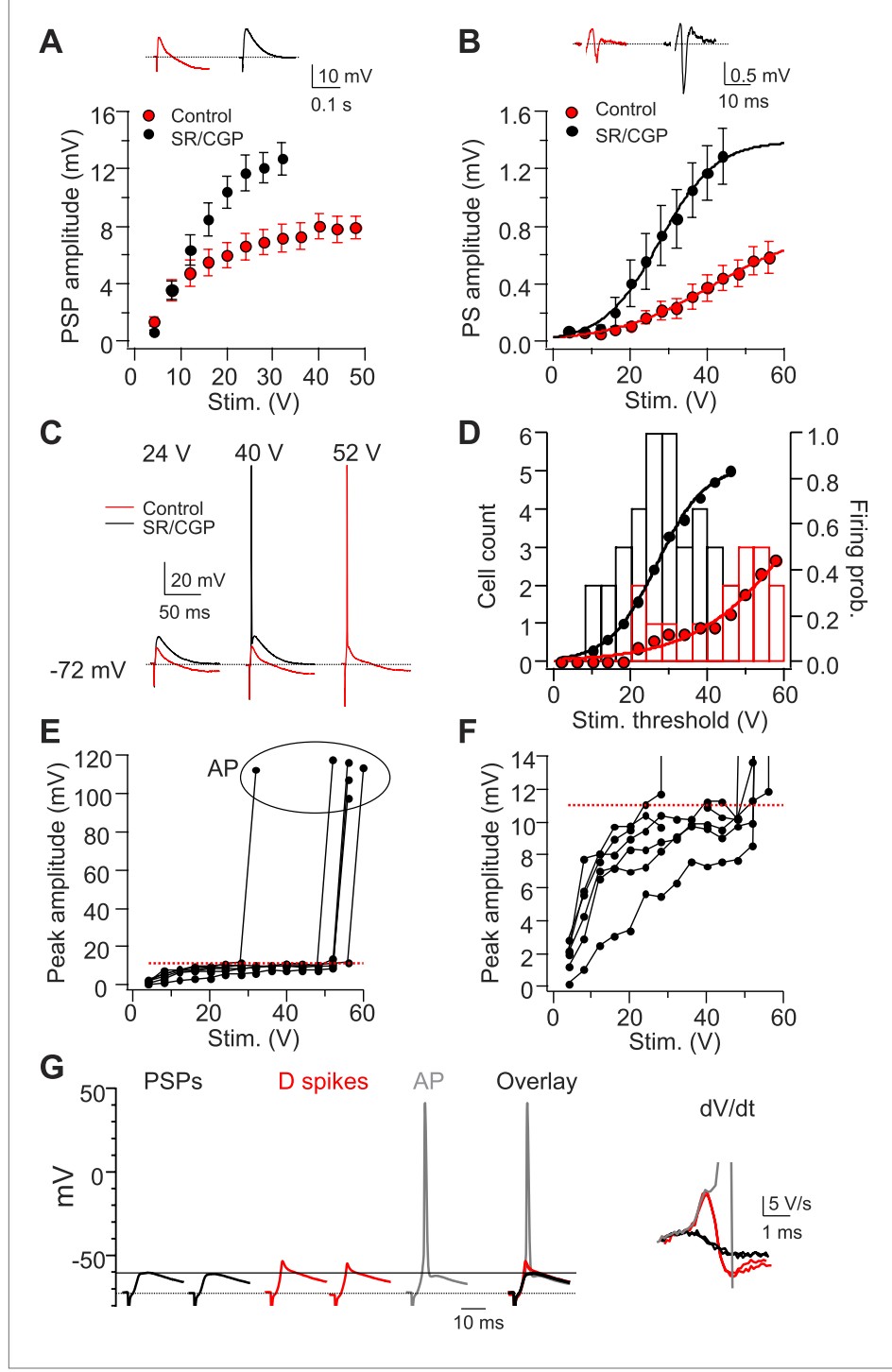

**Figure 9**. Dendritic Na[+] spikes in CA2 PNs overcome inhibition to trigger APs. (**A**) Mean input–output curves of sub-threshold postsynaptic potentials (PSP) in response to a single PP stimulus in the absence (red) or the presence (black) of GABA_A and GABA_B receptor antagonists, 2 µM SR 95531, and 1 µM CGP 55845, respectively (SR/CGP; n = 8–10). Inset: sample traces of PSPs in the absence and the presence of GABAR antagonists. Trials in which stimulus elicited an AP were not included. (**B**) Mean input–output curves of PS in response to a single PP stimulus in the absence or the presence of GABAR antagonists (n = 13). Inset shows sample PS. (**C**) Sample traces of PSPs and APs obtained from somatic whole-cell recordings in response to a PP stimulus with increasing strength in the absence or the presence of GABAR antagonists. (**D**) Bars show population frequency histogram of AP threshold in *Figure 9. Continued on next page*

*Figure 9. Continued*

the absence (red) and the presence (black) of GABAR antagonists. Circles show cumulative distribution of firing probability as function of stimulus voltage (control: red, n = 34 cells; SR/CGP: black, n = 42 cells). (**E**) Peak somatic voltage amplitude plotted against stimulating intensity from individual CA2 PNs which fire APs in responses to a PP stimulus in the absence of GABAR antagonists (n = 6). (**F**) Expanded view of sub-threshold PSP response in (**E**). Note, the red dashed line in (**E**) and (**F**) indicates mean PSP amplitude (10.99 ± 0.59 mV, n = 6) right before CA2 PNs fire APs. (**G**) Left: sample traces of sub-threshold PSPs, dendritic Na$^+$ spikes, and an AP in a CA2 PN in response to single PP stimuli of constant strength near the threshold for AP firing (60 V, 5 trials) in the absence of GABAR antagonists. Note, PSPs are only able to depolarize membrane to −60.4 mV (black line). Right: dV/dt of the corresponding traces shown at left.

The following figure supplement is available for figure 9:

**Figure supplement 1**. Strong PP stimulation with constant strength variably triggers APs, D spikes, or PSPs in the presence of inhibition.

Interestingly, the simultaneous firing of dendritic spikes in six out of twelve branches (~300 μm from the soma) was necessary to evoke a somatic AP (*Figure 10E*). The multiple independent spikes in the dendritic branches were each subject to considerable attenuation as they propagated from the distal region of the dendrites to the primary branch point. However, at primary dendritic branch point, the spikes from each branch summated to produce a very large dendritic spike, which then propagated with little decrement over the short remaining distance to the soma to generate a supra-threshold spikelet (*Figure 10E*). Taken together, these simulations have identified CA2 PN dendritic morphology as a key factor that helps enable the efficient coupling of dendritic Na$^+$ spikes to AP output.

## Discussion

Our results demonstrate that dendritically generated Na$^+$ spikes, driven by cortical inputs to the distal dendrites of CA2 PNs, propagate to the soma and are required to trigger axonal AP output. The dendritic Na$^+$ spikes thus enable the cortical inputs to overcome their unfavorable distal dendritic location and effectively propagate cortical information by shortening spike latency and overcoming powerful inhibition. Our data further show that dendritic Na$^+$ spikes enhance the temporal precision of CA2 PN AP output and are likely crucial for the function of the EC → CA2 → CA1 disynaptic pathway (*Bartesaghi and Gessi, 2004*; *Bartesaghi et al., 2006*; *Chevaleyre and Siegelbaum, 2010*). Given the recent findings that CA2 PNs are critical for social memory (*Hitti and Siegelbaum, 2014*; *Stevenson and Caldwell, 2014*), the dendritic Na$^+$ spikes in these neurons are likely to play a key behavioral role.

### Differential role of dendritic Na$^+$ spikes in enabling neuronal output

The importance of dendritic Na$^+$ spikes in the generation of CA2 PN action potential output represents one end of a continuum of results on the role of these spikes in different classes of neurons. Thus, in both CA1 PNs and neocortical layer 5 neurons, dendritic Na$^+$ spikes normally fail to propagate to the soma and are only weak triggers of somatic APs (*Stuart et al., 1997a*; *Golding and Spruston, 1998*; *Jarsky et al., 2005*; *Larkum et al., 2007*). Nonetheless, dendritic Na$^+$ spikes do sometimes precede somatic spikes and may trigger AP output with strong synaptic stimulation (*Turner et al., 1991*; *Stuart et al., 1997a*; *Golding and Spruston, 1998*) or direct current injection (*Williams and Stuart, 2002*; *Gasparini et al., 2004*).

However, in most neurons, the stimulating intensity required to initiate dendritic Na$^+$ spikes is significantly higher than that required for axonal AP initiation (*Turner et al., 1991*; *Stuart and Sakmann, 1994*; *Stuart et al., 1997a*). Therefore, although dendritic Na$^+$ spikes do have the capability of transforming synaptic inputs into neuronal outputs under certain condition; in most pyramidal neurons, they are neither sufficient nor necessary for axonal AP initiation (*Hausser et al., 2000*; *Spruston, 2008*). Consistent with this notion, we did not detect spikelets (a hallmark of dendritic spikes) in CA1 PN soma in response to a burst of PP stimuli. Instead, our data suggest that temporal summation of somatic depolarization in response to short bursts of PP stimuli, rather than dendritic Na$^+$ spikes, is what drives AP output in CA1 PNs.

A previous study reported that stimulation of PP inputs onto CA1 PNs in rats can evoke dendritic Na$^+$ spikes that sometimes appear as somatic spikelets that may help drive somatic AP output

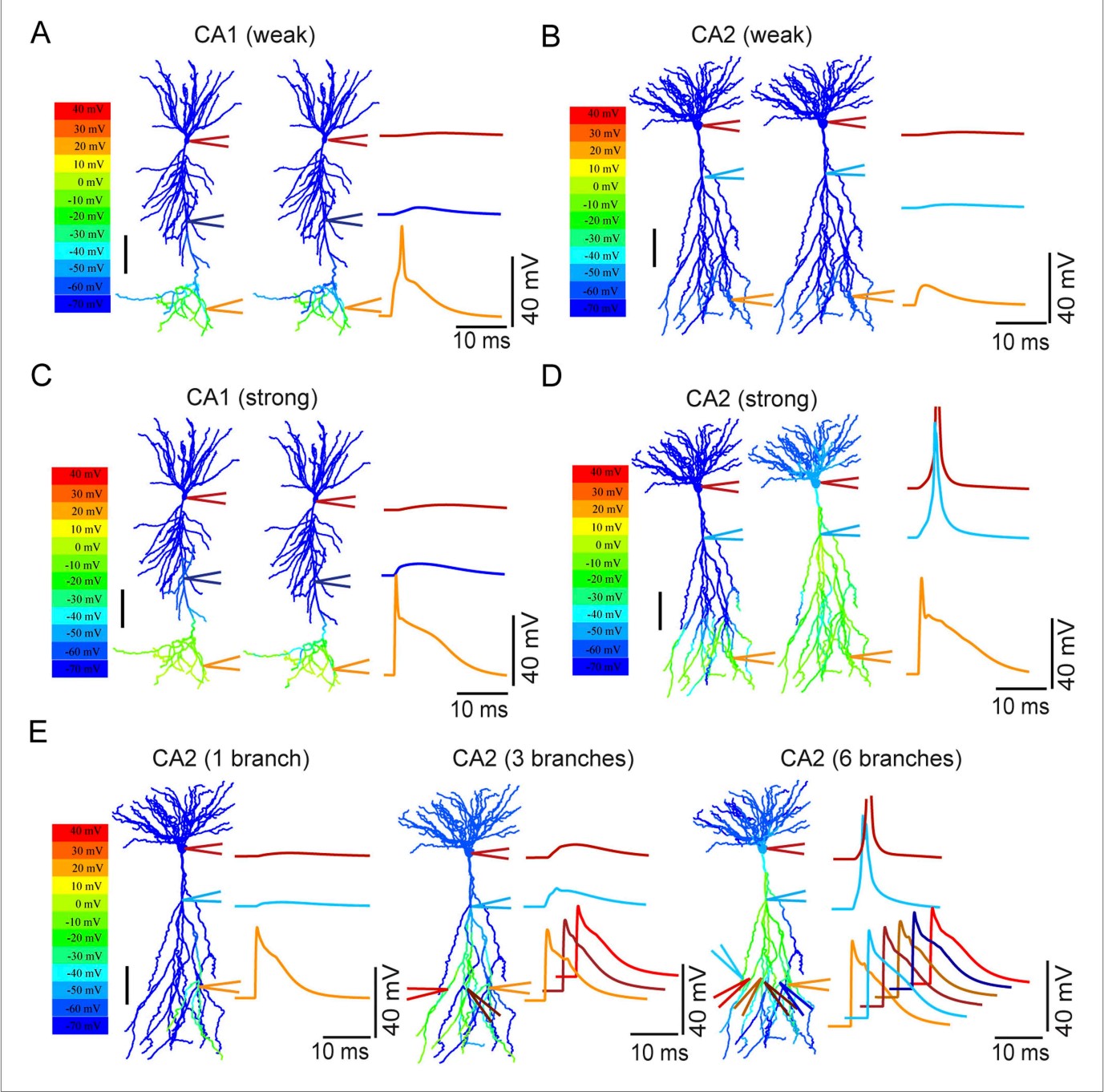

**Figure 10**. Modeling the differential coupling of dendritic Na$^+$ spikes to AP output in CA1 vs CA2 dendrites. (**A**, **B**) Weak PP stimulation (~75 synapses) triggered local spikes at distal apical dendrites in CA1 (**A**), but not in CA2 (**B**), PNs. In both models, dendritic spikes failed to propagate to the soma. (**C**) Strong PP stimulation (~1000 synapses) triggered local Na$^+$ spikes in the apical tuft of the CA1 PN that failed to propagate to the soma. (**D**) Strong PP stimulation (~1000 synapses) triggered local Na$^+$ spikes in the apical tuft of CA2 PN that propagated effectively to the soma and triggered an AP. Note the presence of a prominent dendritic Na$^+$ spike at the CA2 PN primary apical dendrite branch point. APs are truncated. Color maps in (**A**–**D**) represent voltage snapshots. Left: snapshot taken at the time of peak voltage response in distal dendrites of CA1 and CA2 PNs. Right: snapshot taken at the time of peak voltage response at main apical dendritic trunk for CA1 PN or primary dendritic branch point for CA2 PN. Scale: 100 μm. Traces show voltage response at indicated positions. (**E**) Increasing numbers of secondary or tertiary CA2 apical branches (~300 μm from the soma) were activated by ~150 synapses per branch to trigger dendritic spikes. Offset traces at bottom show voltage responses in individual branches. Simultaneous activation of six out of twelve branches triggered a spike at the soma. Note a prominent dendritic Na$^+$ spike at the branch point of primary apical

*Figure 10. Continued on next page*

*Figure 10. Continued*

dendrite of the CA2 PN. Color maps represent snapshots captured at time point of peak voltage observed at the main branch point of CA2 apical dendrites. Scale: 100 µm.

The following figure supplements are available for figure 10:

**Figure supplement 1**. Quantification of dendritic morphology of CA1 vs CA2 using Sholl analysis.

**Figure supplement 2**. Modeling the differential coupling of dendritic Na+ spikes to AP output in CA1 vs CA2 dendrites.

(*Jarsky et al., 2005*). However, such somatic spikelets were only observed in a small minority of cells (~5%) in response to high-frequency bursts of PP stimuli (*Jarsky et al., 2005*), indicating that the vast majority of local spikes generated at the distal apical dendrites failed to propagate to CA1 soma, consistent with our CA1 results. Thus, compared to CA2, the influence of dendritic spikes on AP output in CA1 PNs is rather limited. In contrast to the results on CA1 PNs, dendritic spikes play a more important role in CA1 oriens-alveus interneurons, whose high density of dendritic voltage-gated $Na^+$ channels ensures active spike propagation to the soma, which triggers axonal AP output under some conditions (*Martina et al., 2000*).

## Mechanism for efficient coupling of dendritic Na⁺ spikes to AP output in CA2 PNs

Using a computational model based on reconstructed CA2 and CA1 PNs, we found that the morphology of the CA2 apical dendritic arbor contributes to the efficient coupling of dendritic $Na^+$ spikes to AP output in CA2 PNs. Specifically, the main branch point of the apical dendrites in CA2 acts as a 'hot spot' that integrates spikes from multiple secondary dendrites to generate a large amplitude spike in the short primary dendrite that triggers AP output. This provides a striking example of how dendritic morphology critically influences the propagation of dendritic spikes, as suggested previously (*Vetter et al., 2001*). Our results do not rule out the possibility that additional factors, such as distinct distribution patterns and/or biophysical properties of voltage-gated ion conductances along the CA2 dendrites, may also contribute to the efficient coupling of dendritic spikes to AP output.

## Importance of CA2 dendritic Na⁺ spikes for information propagation through the cortico-hippocampal circuit

To overcome the unfavorable geography of their cortical inputs, CA2 PNs utilize a number of mechanisms that boost the magnitude of the somatic response to the distal synaptic inputs from EC. One set of mechanisms, which remains to be identified, increases the magnitude of the sub-threshold somatic EPSP (*Chevaleyre and Siegelbaum, 2010*). However, despite its larger amplitude, the EPSP generated by the EC inputs is still below the threshold for action potential firing (using somatic current pulses). Given a mean CA2 PN resting potential of −75 mV and an AP threshold of −44 mV, a >30 mV somatic depolarization is required for CA2 PNs to reach the threshold to fire an AP. Yet we find that, in the absence of dendritic $Na^+$ spikes, the EPSP reaches a peak value of around 15–20 mV with strong PP stimulation, far negative to the threshold for eliciting a somatic spike. This is consistent with our finding that dendritic spikes are necessary to trigger an action potential output in response to EC input in the CA2 PNs.

Recent results show that CA2 PNs also receive direct input from DG granule cells through the mossy fiber pathway, although the CA2 PNs lack the thorny excrescences characteristic of CA3 mossy fiber synapses (*Kohara et al., 2014*). However, the DG inputs provide relatively weak synaptic drive onto CA2 PN apical dendrites, evoking small PSPs whose peak amplitude of 5–10 mV is far below the threshold for eliciting spikes with somatic current injection. Nonetheless, the DG inputs can drive CA2 spike output (*Kohara et al., 2014*). To explain the discrepancy between EPSP size and CA2 PN threshold, we suggest that dendritic $Na^+$ spikes may also enable AP output through this additional route of information transfer.

Dendritic $Na^+$ spikes have been suggested to contribute to temporal coding in neural networks (*Ariav et al., 2003*; *Gasparini and Magee, 2006*). Consistent with this idea, we find that CA2 neurons fire precisely and immediately in response to a single PP stimulus. This is in contrast with CA1, where temporal summation of synaptic potentials is required for generating APs. With a burst of PP stimuli,

CA2 fires with the highest probability in response to the first PP stimulus. In contrast, the probability of CA1 firing increases during successive stimuli in the burst. We speculate that this mechanism may exert an influence on the temporal structure of information flow through the cortico-hippocampal circuit. Consistent with our in vitro findings, in vivo extracellular recording demonstrated that CA2 neurons fire APs earlier than CA1 or CA3 neurons in response to EC stimulation (*Bartesaghi and Gessi, 2004*; *Bartesaghi et al., 2006*).

Dendritic $Na^+$ spikes have been observed in vivo in some types of neurons (*Kamondi et al., 1998*; *Waters et al., 2003*; *Smith et al., 2013*). Although it is not known whether dendritic $Na^+$ spikes occur in vivo in CA2 PNs, an active response in the CA2 dendritic field has been observed in response to an electrical stimulus to the PP using extracellular field recording in anesthetized guinea pigs (*Bartesaghi and Gessi, 2004*). Importantly, this active dendritic response precedes that in the cell body layer (*Bartesaghi and Gessi, 2004*), which is consistent with our observations in acute hippocampal slices (*Figure 7*). This suggests that dendritic $Na^+$ spikes in CA2 PNs may similarly influence AP initiation in vivo. Whether cortically driven dendritic $Na^+$ spikes in CA2 PNs occur in awake-behaving animals and whether they are important for specific behaviors, including social behaviors (*DeVito et al., 2009*; *Hitti and Siegelbaum, 2014*; *Pagani et al., 2014*), remain open questions.

## Materials and methods

### Hippocampal slice preparation

Transverse hippocampal slices were prepared from 5- to 8-week old C57BL/6J male mice from the Jackson Laboratory, as described previously (*Chevaleyre and Siegelbaum, 2010*). In brief, animals were anesthetized and killed by decapitation in accordance with institutional regulations. Hippocampi were dissected out, and transverse slices (400 μm thickness) from the dorsal hippocampus were cut on a vibratome (Leica VT1200S, Germany) in ice-cold dissection solution containing (in mM): 10 NaCl, 195 sucrose, 2.5 KCl, 10 glucose, 25 $NaHCO_3$, 1.25 $NaH_2PO_4$, 2 Na Pyruvate, 0.5 $CaCl_2$, and 7 $MgCl_2$. The slices were then incubated in 33°C ACSF (in mM: 125 NaCl, 2.5 KCl, 20 glucose, 25 $NaHCO_3$, 1.25 $NaH_2PO_4$, 2 Na Pyruvate, 2 $CaCl_2$, and 1 $MgCl_2$) for 20–30 min and then kept at room temperature for at least 1.5 hr before transfer to the recording chamber. Cutting and recording solutions were both saturated with 95% $O_2$ and 5% $CO_2$ (pH 7.4). All electrophysiological recording experiments were performed at 31–32°C. For some experiments, a cut was made between CA2 and CA3 regions.

### Somatic and dendritic whole-cell recordings

Whole-cell recordings were obtained from PNs 'blindly' in current clamp mode with a patch pipette (4–6 MΩ for somatic recording; 7–10 MΩ for dendritic recording) containing (in mM): 135 K gluconate, 5 KCl, 0.1 EGTA-Na, 10 HEPES, 2 NaCl, 5 Mg ATP, 0.4 $Na_2$GTP, 10 $Na_2$ phosphocreatine (pH 7.2; 280–290 mOsm). Series resistance and resting membrane potential were monitored throughout each experiment. Neurons with series resistance >25 MΩ (somatic) or >50 MΩ (dendritic) were excluded from analysis. Neurons with resting potential more positive than −60 mV were also rejected from analysis. Synaptic potentials, dendritic spikes, and AP outputs were evoked by monopolar stimulation with a patch pipette filled with 1 M NaCl and located in SLM of the CA1 region (~50 μm from CA2 region). CA2 PNs were identified based on a number of electrophysiological properties as described previously (*Chevaleyre and Siegelbaum, 2010*), including resting membrane potential, input resistance, and firing properties. The paired-pulse ratio was calculated as the ratio of the second to the first EPSP response using two PP stimuli with 50 ms interpulse interval. In some somatic whole-cell recording and all dendritic recording experiments, neurons were filled with biocytin (0.2–1 %, Sigma, St. Louis, MO) during recording and morphological reconstruction was subsequently performed for further verification.

Neurons were held at −70 to −73 mV for input–output curves and for examining the effect of PP stimulation on dendritic spikes and AP output. Resting membrane potential was measured immediately upon break-in. Except for the experiment shown in *Figure 9* and *Figure 9—figure supplement 1*, all EPSPs, dendritic spikes, APs, and population spikes were recorded in the presence of $GABA_A$ and $GABA_B$ antagonists (2 μM SR 95531 and 1 μM CGP 55845, Tocris, Bristol, UK).

### Extracellular field recordings

Extracellular field potentials were recorded with glass patch pipettes containing 1 M NaCl. The recording pipettes were placed in the pyramidal layer or various locations along SR in CA1, CA2, or

CA3 fields. Field responses were evoked using a stimulating electrode placed in SLM of the CA1 field (~50 µm from the border with CA2). Except for the experiments shown in *Figure 9* and *Figure 9— figure supplement 1*, all experiments were performed in the presence of GABAR antagonists. In some experiments, to prevent the contamination of polysynaptic activation from CA2 and CA3 neurons in response to PP stimulation, both CA2 and CA3 regions were cut-off for assessing PS response in CA1 cell body layer.

## Neuronal reconstruction

Neurons were filled with biocytin using whole-cell patch recordings that were held for >15 min to allow for diffusion of biocytin. The slices were fixed and kept overnight in 4% paraformaldehyde in 0.1 M phosphate buffer (PB) at 4°C. The slices were then rinsed five times for five minutes per rinse in 0.1 M PB and were treated with 0.3–1% hydrogen peroxide in 0.1 M PB for 30–40 min. After three rinses, slices were treated with 2% Avidin–Biotin-Peroxidase Complex (ABC, Vector Laboratories, Burlingame, CA) for 1–2 days. Each slice was then developed with 0.05–0.07% 3,3'-diaminobenzidine tetrahydrochloride (DAB) and 0.005% hydrogen peroxide until the slice turned light brown. Subsequently, slices were rinsed in PB several times and processed through increasing concentrations of glycerol and then embedded in mounting media (*Fino and Yuste, 2011*).

Neurons with robust staining of the dendritic tree were reconstructed using Neurolucida software (MBF Bioscience, Williston, VT). The neurons were viewed with a 63x oil objective on a Zeiss upright light microscope. Whole-cell reconstructions included the soma and dendritic branches and shafts, but not dendritic spines.

## Computational modeling

Three dimensional whole neuron reconstructions, including dendritic diameters and lengths, were imported into the NEURON simulation environment (*Hines and Carnevale, 1997*). To build active models of CA1 and CA2 PNs, we used a similar approach as described previously (*Jarsky et al., 2005*). To the best of our knowledge, there are no available experimental data regarding distributions or biophysical properties of dendritic voltage-gated conductances from mouse hippocampal CA1 or CA2 PNs. Thus the parameters used in our models were derived from the experimental data obtained from rat hippocampal CA1 PNs. The models incorporated passive membrane properties ($R_m$ = 40,000 $\Omega$ cm$^2$, $C_m$ = 0.75 µF/cm$^2$, $R_i$ = 150 $\Omega$ cm). To account for spines, $C_m$ of the dendritic compartments was multiplied by a spine scale factor and their $R_m$ was divided by the same factor. In the CA1 PN model, we used a spine scale factor of 2 in compartments 150 µm beyond the soma, whereas in the CA2 PN model we used spine scale factors of 2 or 3 for compartments below or beyond 150 µm from the soma, respectively. These spine scale factor values were chosen to match the membrane time constant and input resistance values of the models to our experimental values.

The models also included four active conductances: a Na$^+$ conductance ($G_{Na}$), a delayed rectifier K$^+$ conductance ($G_{Kdr}$), an A-type K$^+$ conductance ($G_{KA}$), and a hyperpolarization-activated cation conductance ($G_h$). The biophysical parameters of $G_{Na}$, $G_{Kdr}$, $G_{KA}$, and $G_h$ were implemented, as described previously (*Magee, 1998*; *Jarsky et al., 2005*). These conductances were inserted in all compartments of the models. The distribution of $G_{Na}$ and $G_{Kdr}$ is uniform throughout the somato-dendritic axis in both CA1 and CA2 models with a conductance value of 0.022 S/cm$^2$ and 0.035 S/cm$^2$ respectively. $G_{KA}$ was modeled with sixfold increase in conductance along the somato-dendrtic axis as described previously (*Hoffman et al., 1997*; *Jarsky et al., 2005*), with conductance values of 0.035 S/cm$^2$ at the soma in both models. $G_h$ was modeled with a sevenfold increase in conductance along the somato-dendritic axis as described previously for CA1 PNs (*Magee, 1998*), whereas in CA2 PNs its distribution was uniform based on our inspection of immunocytochemistry results (*Santoro et al., 2004*). All simulations were performed at a resting potential of −70 mV.

All excitatory synapses were modeled using two exponential functions to describe the conductance time course ($\tau_{rise}$ of 0.2 ms and $\tau_{decay}$ of 2 ms) with a reversal potential of 0 mV with specific synaptic conductance values described below. For distal synaptic activation, excitatory synapses (0.0002 µS per synapse) were distributed randomly onto the distal dendritic arbor (>400 µm from the soma). Weak synaptic stimulation was performed by activating ~75 synapses randomly, whereas in strong synaptic stimulation ~1000 synapses were activated randomly in both models. For the branch model (*Figure 10E*), each branch (~300 µm from the soma) was activated by ~150 synapses.

## Data acquisition and analysis

Data were digitized with a Digidata 1440A interface (Molecular Devices, Sunnyvale, CA) and were acquired using AxoGraph X software (AxoGraph, Berkeley, CA). Data analysis was performed using Igor Pro (Wavemetrics, Lake Oswego, OR), AxoGraph X, and Excel (Microsoft, Redmond, WA). Phase-plane plot and dV/dt values were obtained with the build-in programs in AxoGraph X. AP threshold was defined as the somatic voltage at which dV/dt exceeded 10 V/s or 50 V/s. To determine the EPSP amplitude when dendritic spikes were present (e.g. *Figure 2—figure supplement 1A,B*), EPSP amplitude was determined at the peak depolarization of the EPSP waveform (the latency of peak EPSP is typically >8 ms) following the dendritic spike (dendritic spike latency is usually <5 ms). For *Figure 1C*, EPSP data were excluded if a cell started to fire somatic APs in response to increasing stimulation intensity. In a subset of CA2 PNs, PP stimulation with constant high-intensity stimuli variably triggered full-blown APs, dendritic spikes without APs, or PSPs in the presence of inhibition (e.g. *Figure 9G*, *Figure 9—figure supplement 1*). In those cases, the PSP values were used to generate the input–output relation of PSP and stimulating intensity (*Figure 9A*). Statistical comparisons were performed using Student's t test or ANOVA. Results are expressed as mean ± SEM.

## Acknowledgements

We thank Pablo Ariel, Randy Bruno, Attila Losonczy, and Bina Santoro for critically reading the manuscript and for helpful discussions. This work was supported by the Howard Hughes Medical Institute and grant R01MH104602 from NIH.

## Additional information

### Funding

| Funder | Grant reference number | Author |
| --- | --- | --- |
| Howard Hughes Medical Institute | | Steven A Siegelbaum |
| National Institute of Mental Health | R01MH104602-01 | Steven A Siegelbaum |

The funders had no role in study design, data collection and interpretation, or the decision to submit the work for publication.

### Author contributions

QS, Designed the research, Performed all experiments, Analyzed the data, Drafted the paper; KVS, Performed the computational modeling; AS, Performed Sholl analysis; SAS, Contributed to the experimental design, Discussed the data, Drafted the paper

### Ethics

Animal experimentation: This study was performed in strict accordance with the recommendations in the Guide for the Care and Use of Laboratory Animals of the National Institutes of Health. All mouse studies were approved by the Institutional Animal Care and Use Committee (IACUC) of Columbia University (protocol Number AC-AAAF6104). Animal housing, husbandry, and euthanasia were conducted under the guidelines of the Institute of Comparative Medicine, Columbia University.

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
