## [Decision Letter]

Thank you for sending your work entitled “Dendritic Na^+^ spikes enable cortical input to drive action potential output from hippocampal CA2 pyramidal neurons” for consideration at *eLife*. Your article has been favorably evaluated by Eve Marder (Senior editor), Gary Westbrook (Reviewing editor), and 2 reviewers.

The Reviewing editor and the reviewers discussed their comments before we reached a decision, and assembled the following comments to help you prepare a revised submission. The reviewers found the experiments to be convincing and the findings to be very significant, demonstrating a pyramidal neuron with reliably propagating D spikes. However, both reviewers had concerns about the conclusions regarding morphology and aspects of the modeling that require your attention.

Major comments:

1) Modeling. The current work leaves little doubt that distal PP stimulation results in dendritic Na^+^ spiking that can facilitate somatic action potential initiation including in physiological-like conditions (even in the presence of GABA-mediated inhibition). However, the authors' conjecture that morphology alone seems sufficient to explain efficiently propagation of dendritic spikes in CA2 PCs is rather simplistic and relatively untested. Although the authors attribute the differences between CA1 and CA2 cells to morphology, they actually have differences in the conductances in the two models as well. The smaller A-type K conductance in the CA2 model could be a major factor contributing to differences in D spike propagation between the models of the two cell types. To test this idea rigorously, the authors should put the CA1 conductances into the CA2 cell, and vice versa. If morphology is truly the main difference, as the authors claim, it should still be true that the chimeric model shows better D spike propagation in CA2 than in CA1, despite the reversal of the conductances in the two models.

2) The authors do not show convincingly that the morphological differences between CA1 and CA2 that are apparent in the model are truly representative. To make this conclusion, quantitative data on the morphological properties of the dendritic trees of these two cell types are needed, as opposed to the anecdotal data shown here (one reconstructed neuron of each type). These data, on their own, could bolster the case that morphological differences are real and therefore could account for the differences in D spike propagation. Using modeling to explore more deeply how quantitative morphological differences contribute to D spike initiation/propagation would also strengthen the paper.

3) The authors' simulations indicate that distributed synaptic input across multiple secondary branches of the CA2 dendrite (the proposed “unique morphological feature” of CA2 PCs) allows for independent Na^+^ spikes to be generated in these processes, resulting in a large summated response at the primary dendrite which can then effectively propagate to the soma and initiate action potential firing. However, a more detailed analysis is needed. Specifically, in the simulations, are D spikes generated more readily in the distal dendrites of CA2 compared to CA1, or do they propagate better? One possibility is that the major branch point of the main apical dendrite is closer to the soma in CA2. If this is where D spikes fail, they could be larger in CA2 simply because that location is closer to the soma. Is that what's happening or is there some other explanation?

4) The paragraph discussing the previous work in CA1 neurons (20) misses an important point. The fact that only a small minority of CA1 cells exhibit somatically recorded spikelets likely reflects the fact that D spikes fail so far from the soma that a spikelet is not visible in somatic recordings, rather than reflecting the absence of a D spike in response to PP stimulation. The 2005 study used modeling to predict this result. Those authors also predicted that coincident stimulation of the Schaffer collaterals could facilitate forward propagation of D spikes generated in the tuft. They then tested this prediction experimentally and found that it was indeed the case. Thus, there is no need to invoke a difference between rats and mice to explain the lack of spikelet observed in CA1 in the present study. Rather, it is likely the D spikes are also generated in CA1 tuft dendrites, but their failure to propagate reliably results in so much attenuation that the D spikes are completely invisible in the soma. This point should be discussed.

5) The authors imply that dendritic Na^+^ spikes are a requirement for PP-mediated somatic spiking as, following dendrite-targeted TTX application, PP stimulation no longer evoked somatic action potentials. In a few places, the authors claim that D spikes are “necessary” for firing in response to stimulation of the PP. However, this seems like a rather sweeping conclusion given the limited conditions used to stimulate PP inputs. Other conditions not tested in the experiments (e.g. in vivo) could lead to PP-mediated action potentials even in the absence of D spikes. This is easily corrected with some wording changes. For example, rather than saying, “Thus, cortically-driven dendritic Na^+^ spikes are necessary for AP output in CA2 PNs to either single or bursts of PP input.”, they should say, “Thus, cortically-driven dendritic Na+ spikes were necessary for AP output in CA2 PNs to either single or bursts of PP input.” The difference is subtle. The latter sentence refers only to the experiments the authors did, while the former suggests that it would always be true. Additional discussion could also clarify this point.

---

## [Author Response]

*1) Modeling. The current work leaves little doubt that distal PP stimulation results in dendritic Na*^*+*^
*spiking that can facilitate somatic action potential initiation including in physiological-like conditions (even in the presence of GABA-mediated inhibition). However, the authors' conjecture that morphology alone seems sufficient to explain efficiently propagation of dendritic spikes in CA2 PCs is rather simplistic and relatively untested. Although the authors attribute the differences between CA1 and CA2 cells to morphology, they actually have differences in the conductances in the two models as well. The smaller A-type K conductance in the CA2 model could be a major factor contributing to differences in D spike propagation between the models of the two cell types. To test this idea rigorously, the authors should put the CA1 conductances into the CA2 cell, and vice versa. If morphology is truly the main difference, as the authors claim, it should still be true that the chimeric model shows better D spike propagation in CA2 than in CA1, despite the reversal of the conductances in the two models*.

The reviewers raise an important point. Based on the reviewers’ comment, we asked whether it is possible to explain differences in CA2 versus CA1 excitability without postulating a difference in A-type conductance. We now find that, with slight adjustments in the models’ voltage-gated conductance parameters, we can reproduce our experimental results based solely on morphological differences, using an identical value of A-type conductance – as well as identical Na^+^ and delayed rectifier K^+^ conductances – in the CA2 and CA1 PNs. We present these new results in Figure 10 and new Figure 10—figure supplement 2.

*2) The authors do not show convincingly that the morphological differences between CA1 and CA2 that are apparent in the model are truly representative. To make this conclusion, quantitative data on the morphological properties of the dendritic trees of these two cell types are needed, as opposed to the anecdotal data shown here (one reconstructed neuron of each type). These data, on their own, could bolster the case that morphological differences are real and therefore could account for the differences in D spike propagation. Using modeling to explore more deeply how quantitative morphological differences contribute to D spike initiation/propagation would also strengthen the paper*.

We now have included modeling results from two more reconstructed CA1 and CA2 PNs (new Figure 10—figure supplement 2), and performed Sholl analysis to show that the branching patterns of CA2 apical dendritic arbors are quantitatively different from their CA1 counterparts (new Figure 10—figure supplement 1). Specifically, Sholl analysis demonstrated CA2 PNs have significantly more second and tertiary dendritic branches extending toward SLM than do CA1 PNs. Furthermore, we have performed a second set of computational modeling using the same voltage-gated conductance parameters used in the original pair of reconstructed CA1 and CA2 PNs in two different newly reconstructed CA1 and CA2 PNs and we obtained similar results as in our original models (new Figure 10—figure supplement 2). Together, we believe that our new data significantly strengthen one of our main conclusions that dendritic geometry of CA2 PNs contributes to the efficient coupling of dendritic spikes to AP output.

*3) The authors' simulations indicate that distributed synaptic input across multiple secondary branches of the CA2 dendrite (the proposed “unique morphological feature” of CA2 PCs) allows for independent Na*^*+*^
*spikes to be generated in these processes, resulting in a large summated response at the primary dendrite which can then effectively propagate to the soma and initiate action potential firing. However, a more detailed analysis is needed. Specifically, in the simulations, are D spikes generated more readily in the distal dendrites of CA2 compared to CA1, or do they propagate better? One possibility is that the major branch point of the main apical dendrite is closer to the soma in CA2. If this is where D spikes fail, they could be larger in CA2 simply because that location is closer to the soma. Is that what's happening or is there some other explanation?*

We appreciate the reviewers raising this issue. Our data suggest that, compared to CA1, the better propagation due to dendritic geometry of CA2, rather than more excitable dendrites, are important for coupling dendritic spikes to AP output. Our experimental dendritic whole cell current-clamp recordings show the current threshold to generate local spikes is comparable between CA1 and CA2 (CA1: 1.6 ± 0.2 nA versus CA2: 1.7 ± 0.2 nA), indicating that the excitability of CA2 dendrites is comparable to CA1. This point is further supported by our models, in which CA1 distal dendrites can readily generate local spikes (Figure 10, new Figure 10—figure supplement 2). To allow direct visualization of dendritic spike propagation, we now have performed further analyses in both CA1 and CA2 models (Figure 10, new Figure 10—figure supplement 2, see the model session in Results for details). In summary, prominent dendritic spikes (>30 mV in amplitude) are visible at the main branch point of the apical dendrite in CA2 PNs in response to EC inputs. Due to the proximity of this branch point to the soma, the dendritic spikes are subject to very little additional attenuation such that large spikelets can be readily detected at the soma in our experimental conditions. Thus, our models support the idea that the main branch point of CA2 PNs acts as a “hot spot” zone to initiate dendritic spikes and trigger axonal AP output. We now have discussed this point in the revised manuscript.

*4) The paragraph discussing the previous work in CA1 neurons (*[20]*) misses an important point. The fact that only a small minority of CA1 cells exhibit somatically recorded spikelets likely reflects the fact that D spikes fail so far from the soma that a spikelet is not visible in somatic recordings, rather than reflecting the absence of a D spike in response to PP stimulation. The 2005 study used modeling to predict this result. Those authors also predicted that coincident stimulation of the Schaffer collaterals could facilitate forward propagation of D spikes generated in the tuft. They then tested this prediction experimentally and found that it was indeed the case. Thus, there is no need to invoke a difference between rats and mice to explain the lack of spikelet observed in CA1 in the present study. Rather, it is likely the D spikes are also generated in CA1 tuft dendrites, but their failure to propagate reliably results in so much attenuation that the D spikes are completely invisible in the soma. This point should be discussed*.

We agree with the reviewers. Our experimental and theoretical results confirm that dendritic spikes are generated in distal CA1 dendrites but are severely attenuated at the soma. We now discuss this important point more clearly.

*5) The authors imply that dendritic Na+ spikes are a requirement for PP-mediated somatic spiking as, following dendrite-targeted TTX application, PP stimulation no longer evoked somatic action potentials. In a few places, the authors claim that D spikes are “necessary” for firing in response to stimulation of the PP. However, this seems like a rather sweeping conclusion given the limited conditions used to stimulate PP inputs. Other conditions not tested in the experiments (e.g. in vivo) could lead to PP-mediated action potentials even in the absence of D spikes. This is easily corrected with some wording changes. For example, rather than saying, “Thus, cortically-driven dendritic Na*^*+*^
*spikes are necessary for AP output in CA2 PNs to either single or bursts of PP input.”, they should say, “Thus, cortically-driven dendritic Na*^*+*^
*spikes were necessary for AP output in CA2 PNs to either single or bursts of PP input.” The difference is subtle. The latter sentence refers only to the experiments the authors did, while the former suggests that it would always be true. Additional discussion could also clarify this point*.

We agree that the necessity of dentritic spikes for AP output in CA2 holds true only under our experimental conditions, and thus have clarified this point based on the reviewers’ comment.